# Applying a transformer architecture to intraoperative temporal dynamics improves the prediction of postoperative delirium

Niklas Giesa [1] ✉, Maria Sekutowicz[1,2,3], Kerstin Rubarth[1], Claudia Doris Spies[2], Felix Balzer [1], Stefan Haufe[1,4,5,6,7] & Sebastian Daniel Boie [1,7]

## Abstract

**Background** Patients who experienced postoperative delirium (POD) are at higher risk of poor outcomes like dementia or death. Previous machine learning models predicting POD mostly relied on time-aggregated features. We aimed to assess the potential of temporal patterns in clinical parameters during surgeries to predict POD.

**Methods** Long short-term memory (LSTM) and transformer models, directly consuming time series, were compared to multi-layer perceptrons (MLPs) trained on time-aggregated features. We also fitted hybrid models, fusing either LSTM or transformer models with MLPs. Univariate Spearman's rank correlations and linear mixed-effect models establish the importance of individual features that we compared to transformers' attention weights.

**Results** Best performance is achieved by a transformer architecture ingesting 30 min of intraoperative parameter sequences. Systolic invasive blood pressure and given opioids mark the most important input variables, in line with univariate feature importances.

**Conclusions** Intraoperative temporal dynamics of clinical parameters, exploited by a transformer architecture named TRAPOD, are critical for the accurate prediction of POD.

## Plain Language Summary

Delirium manifests as confusion and a lack of awareness. Postoperative delirium is a severe medical complication that can occur after surgery. Currently, there is no specialized medical treatment available, but early detection can be useful to implement preventative measures. In this study, we applied various computational models to clinical data such as repeated blood pressure recordings. Data recorded during the first half of surgeries were most predictive for postoperative delirium. This information could be used to better focus preventative measures after surgery, such as transferring vulnerable patients to quieter wards facilitating recovery.

Postoperative delirium (POD), described as an acute brain dysfunction[1], can lead to adverse outcomes such as a prolonged length of hospital stay, dementia, or death[1–7]. Common symptoms include disturbances in awareness, attention, or cognition[8]. Although POD may occur early after surgery in the recovery room, where patients are under special surveillance, it is often under-diagnosed[1,9]. This can be explained by the key feature of delirium, namely the fluctuation of multi-factorial symptoms[1,2]. Studies have reported prevalence rates ranging from 5% to 52%, contingent upon the patient cohort and inclusion criteria utilized[10,11]. In the recovery room setting, POD can be diagnosed using the nursing delirium screening scale (Nu-DESC)[12] which has been shown to achieve high sensitivities ranging

between 90 and 95%[2]. In contrast to early POD screening with the Nu-DESC, the confusion assessment method (CAM-ICU) has prevailed in intensive care units (ICUs) as a valid clinical diagnostic test[13]. Although not all predisposing features have been unraveled yet, clinical guidelines suggest POD preventive measures. These include early detection, the reduction of surgical stress, the active adjustment of the sedation depth, and enhanced recovery programs[2,14].

Over the past years, researchers have developed and validated prognostic models in order to provide an early risk evaluation before POD onset[15,16]. Respective models have been rarely implemented into clinical practice, though[15–17]. Most of the previous studies used logistic regression (LR) models

[1]Charité - Universitätsmedizin Berlin, Institute of Medical Informatics, 10117 Berlin, Germany. [2]Charité - Universitätsmedizin Berlin, Department of Anesthesiology and Operative Intensive Care Medicine (CCM, CVK), 13353 Berlin, Germany. [3]Berlin Institute of Health at Charité - Universitätsmedizin Berlin, BIH Biomedical Innovation Academy, 10117 Berlin, Germany. [4]Charité - Universitätsmedizin Berlin, Brain and Data Science Group at Berlin Center for Advanced Neuroimaging (BCAN), 10117 Berlin, Germany. [5]Technische Universität Berlin, Head of Uncertainty, Inverse Modeling and Machine Learning (UNIML), 10587 Berlin, Germany. [6]Physikalisch-Technische Bundesanstalt (PTB), Working Group 8.44 Machine Learning and Uncertainty, 10587 Berlin, Germany. [7]These authors contributed equally: Stefan Haufe, Sebastian Daniel Boie. ✉e-mail: niklas.giesa@charite.de

with predisposing risk factors as inputs and CAM-ICU scores as labels to be predicted[15,16,18–24]. Temporal signals in feature time series were commonly aggregated with summary statistics arranged into tabular datasets[15].

Wassenaar et al. and Boogard et al. trained LR models with min- or max-aggregated intensive care data, like vital signs, recorded within 24 h after admission[21,22]. Kim et al. fitted a LR with static features, like age or comorbidities, as well as mean values from laboratory tests to predict POD[23]. Zucchelli et al. and Vreeswijk et al. developed delirium scores based on a LR approach using static data from a patient's medical history and counts of given medications[24,25]. Xing et al. explicitly trained a POD prognostic model with perioperative tabular data like type of anesthesia, surgery duration, or last intraoperative lab test results[26]. Apart from these authors, only a few others published implementation details allowing external validation[21–24,26].

Multiple studies used non-linear machine learning (ML) algorithms, primarily tree-based (TB) models, trained on more complex clinical feature sets[18,19,27–30]. Bishara et al.[29] developed a prognostic XGBoost[31] TB model with 85.1% AUROC (area under receiver operating characteristic curve) and a precision of 14.4% at 80.6% recall[32]. Their model outperformed multi-layer perceptron (MLP)[33] networks and ingested preoperative data labeled with the Nu-DESC[29]. In addition to TB approaches, Bhattacharyya et al.[18] and Liu et al.[30] trained deep learning (DL) models using long short-term memory (LSTM)[34] networks. These networks are generally capable of learning temporal dynamics by sequentially ingesting time series[35]. Instead of focusing on POD, the two teams predicted delirium onset as assessed by the CAM-ICU during ICU stays[18,30]. The best performing LSTM by Bhattacharyya et al. used data retrieved from a 24 h observation window and predicted over 12 h. The achieved performance was 88.39% AUROC and 34.97% AUPRC (area under precision recall curve)[18]. Liu et al. combined a TB approach with a LSTM method and achieved 95.2% AUROC and 75.9% AUPRC. Their model predicted over 6 h with data from a 12 h window after hospital admission[30]. None of the models were made openly accessible.

The current development of large language models (LLMs)[36] has been driven by the invention of the transformer (TRAN) architecture, which is based on the attention mechanism[37]. Attention increases efficiency by processing entire time series in parallel instead of treating inputs sequentially[37,38]. While LSTMs have been used as prognostic models in a vast number of studies, the application of TRAN models to clinical time series is a relatively new field of research[39]. Guo et al. and Peng et al. used the attention mechanism to represent temporal dependencies between clinical events[40,41]. Bednarski et al. combined a TRAN- with a convolutional neural network (CNN) to learn spatio-temporal dependencies in 24 h clinical time series data[42]. Che et al.[43] enhanced a gated recurrent unit (GRU) architecture for learning temporal missingness patterns while predicting a clinical target variable. Missing observations in time series and data heterogeneity have, however, been identified as impeding temporal representation in complex DL models[44].

To best of our knowledge, no previous work has considered intraoprative temporal dynamics for the POD prediction task. The utility of time series models, such as LSTM and TRAN, is currently under-explored in this field of clinical research. Information from temporal changes of subsequently measured parameters, such as vital signs, are lost in summary statistics. We hypothesized that the imminent temporal patterns of perioperatively acquired measures might contain information about the future POD onset. Characteristic temporal relationships between clinical parameters as well as the presence or absence of parameter values might play a crucial role for the POD prediction task.

In the present study, we evaluate whether temporal progression and the presence or absence of perioperative parameters provide added value over feature aggregations. We compare the performance of a variety of ML architectures using either summary statistics, entire time series, or combinations of both as input features. Presence of POD is labeled with the Nu-DESC. Specific architectures include TRAN models, LSTM networks, MLPs, and combinations thereof. ML analyses are complemented by univariate analyses, which enables us to analyze distinct feature effects on POD discriminability.

As a result, a tailored TRAN architecture performs best acting on the first 30 min of intraoperative time series. The temporal courses of clinical measurements, such as invasive systolic blood pressure or the application of the opiod remifentanil, have high predictive potential. We show possible model configurations for making predictions in clincal practice.

## Methods
### Ethics approval
This work was approved by the local institutional review board of the Charité – Universitätsmedizin Berlin, 10117 Berlin, Germany under the internal ethics approval number EA4/254/21. Patients gave general consent to share data for future research purposes at our institution via signing the hospital treatment contract. Regarding our specific study, the IRB EA4/254/21 covered the retrospective processing of de-identified data records.

### Target variable definition
Occurrence of POD was defined with the Nu-DESC score[45]. Nu-DESC assessment in the recovery room for POD diagnosis has been established as a standard procedure at our clinical institution over the past years. The Nu-DESC test assigns 0–2 points to each of the following five categories: disorientation, inappropriate behavior, inappropriate communication, hallucinations, and psycho-motor retardation[46]. The Nu-DESC score is the sum of all points across categories. If a patient was assessed with at least one Nu-DESC point, the corresponding surgery was labeled POD positive. If all Nu-DESC categories were scored zero, the surgery was labeled POD negative. We applied a threshold of one, as suggested by clinical guidelines[2,14], due to the resulting enhancement in sensitivity, aiming at detecting highly vulnerable patients. Our target variable indicating the presence of POD is thus defined as:

$$Y = \begin{cases} 1, & \text{at least one Nu-DESC score} \geq 1 \\ 0, & \text{all Nu-DESC scores} = 0 \end{cases} \tag{1}$$

### Perioperative timing
We outlined important events and time intervals occurring within the perioperative context (see Fig. 1). Anesthesia is induced as a preparation for surgical procedures including mechanical ventilation. Within minutes after anesthesia termination (including extubation) in the operating room, the

**Fig. 1 | Perioperative time phases and events.** Hospital admission and discharge are not shown. The beginning and the end of the intraoperative phase are denoted by T_begin and T_end, respectively.

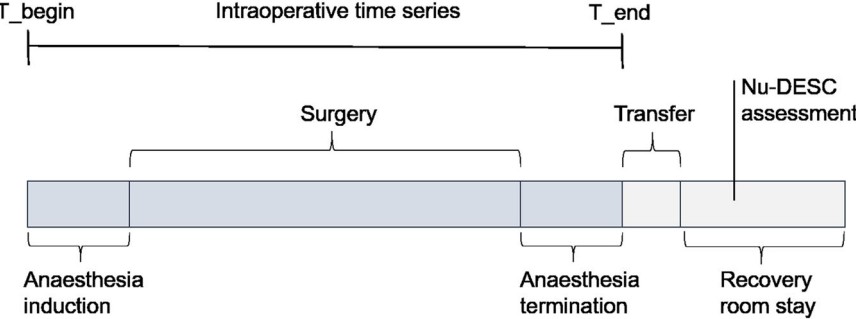

**Table 1 | Descriptions of extracted features**

| Clinical domain | Time variance | Data type | Example | #Features |
|---|---|---|---|---|
| Anesthesia type | Static | Binary | Epidural (anes_epidu) | 18 |
| Demographics | Static | Numeric | Body weight (body_weight) | 6 |
| EEG | Dynamic | Numeric | Sedation depth (psi) | 5 |
| Drug types | Dynamic | Binary | Antibiotics (in_class_antibiotics) | 5 |
| Inputs | Dynamic | Numeric | Amount propofol (in_am_propo) | 19 |
| Outputs | Dynamic | Numeric | Urine output (out_vol_urine) | 3 |
| Laboratory | Dynamic | Numeric | Blood sodium (lab_sodium) | 8 |
| Medical history | Static | Numeric | Heart failure (hst_hfail) | 33 |
| Operation history | Static | Binary | Abdominal (hst_digest_op) | 11 |
| Scores | Static | Numeric | ASA classificaton (asa) | 4 |
| Life-support | Dynamic | Binary | Endotrachial ventilation (durat_endotr) | 16 |
| Vital signs | Dynamic | Numeric | Invasive sytolic blood pressure (ibp_sys) | 12 |
| Respiratory | Dynamic | Numeric | Tidal volume (tid_vol) | 8 |
| | | | | 148 |

Clinical domain, encoding procedure, examples, and the number of features per domain are outlined. 148 features were selected from the clinical source systems. Four composite features – e.g., combining set and measured respiratory rate – and 67 binary missing indicator variables were added. We calculated the cumulative sum for amounts and volumes over time for 19 medications. The final feature set comprised 238 features.

patient is transferred to the recovery room. Right after the recovery room admission, caretakers apply the Nu-DESC test for assessing delirium. Prior to the surgery, an anesthesia consent meeting is conducted, in which the patient is informed about procedures and risks regarding the surgery. Static preoperative characteristics, such as demographic variables, are recorded during this meeting.

We defined the intraoperative phase from the beginning of the anesthesia induction (T_begin) to the end of the anesthesia termination (T_end). We divided the intraoperative phase into different observation windows. One observation window [in minutes] is denoted as $W = [T_x, T_y]$ where $T\_begin \leq T_x < T_y \leq T\_end$. The window endpoints $T_x, T_y$ were defined as time deltas [in minutes] either relative to T_begin (no prefix) or to T_end (negative prefix). The largest considered interval was the full intraoperative interval between T_begin or T_end. Example windows are listed in Supplementary Table 1.

**Feature selection**

We identified promising features due to a literature review and found a potential number of 197 features in the clinical information systems (CIS) across three different hospital sites of our center (see Supplementary Data 1). We selected 148 out of 197 variables due to their availability for at least 1% of patients. Thus, we investigated the influence of rare as well as highly (100%) available features. Details on feature availability (and missingness) are provided in Supplementary Data 2.

Table 1 summarizes the feature encoding process. Feature values were either considered as time-static, not changing over the intraoperative phase, or time-dynamic, fluctuating during the surgery. In addition to 148 selected features, we derived four composite features that combined 1. non-invasive and invasive mean blood pressure, 2. set and measured fraction of inspired oxygen (FiO2), 3. invasive and spontaneous urine output, 4. set and measured positive end-expiratory pressure (PEEP). Single feature vectors were simply concatenated for these pooled measures before sampling with an interval of e.g. three minutes. We introduced four composite features to increase data availability for these variables, as they depict the same physiological attributes such as blood pressure. By keeping the original single vectors in our feature set, we could differentiate e.g. between spontaneous and mechanical ventilation.

For 19 medications, the cumulative sum of administered volumes or amounts over time was calculated. In addition to these derived variables, we encoded data availability with binary missingness indicators for 67 features, assigning 1 if a value was missing and 0 otherwise[47]. Binary missingness

indicators were included for the following clinical domains: EEG (5 features), inputs (19 features), outputs (3 features), laboratory values (8 features), scores (4 features), vital signs (12 features), respiratory signals (8 features), demographics (5 features excluding gender), and four composite features. For other domains, like the medical history, we could not differentiate between a missing measurement (variable not present) and a true negative (variable encodes a negative result). Thus, no binary missingness indicators were added here. The patient's medical history was encoded by ICD codes, surgical procedures were extracted via the German Operationen und Prozeduren Schlüssel (OPS)[48] (see Supplementary Data 3). A total of 238 features were included into our final feature set (see Supplementary Data 4).

**Data preprocessing**

Extraction scripts were visually validated on 100 subjects by testing that the extracted data resulted in the same temporal sequences in the CIS front-end. Some features contained multiple outliers that were not physiologically possible, like negative blood pressure. To clean the extracted data, we defined valid ranges consulting physicians who work in anesthesiology (see Supplementary Data 1). If feature values did not fall into these ranges, we set the corresponding value to NaN.

We explored original sampling intervals for high-frequency time-dynamic vital signs. Supplementary Fig. 1 plots the distribution of time deltas between the occurrence of two consecutive values. For these vital signs, we observed a peak in deltas around 3 min. Additionally we wanted to investigate a larger sampling interval. Thus, we sampled our time-dynamic features with three and five minutes intervals applying mean aggregation (see Supplementary Methods 1, Supplementary Fig. 2). Missing values in feature time series were imputed using LOCF approach[49]. Mean imputation was used for remaining sequential values and for tabular data. Statistics for the imputation process were always calculated on the training sets.

To evaluate MLP models with different types of aggregated tabular data, we calculated the 10th, 50th, and 90th percentiles for time-dynamic features. In addition, we calculated cumulated sums of administered volumes and amounts of each medication across time. This type of aggregated tabular data was denoted as TAB_P. To compactly summarize temporal dynamics in the time series, we derived the Hjorth parameters[50] complexity, activity, and mobility as well as Haar wavelet coefficients[51]. We called this type of aggregation TAB_F. For the third tabular set, we vectorized the feature values measured at different time steps, e.g., the blood pressure (BP) measured every 3 min within a 30 min window was converted

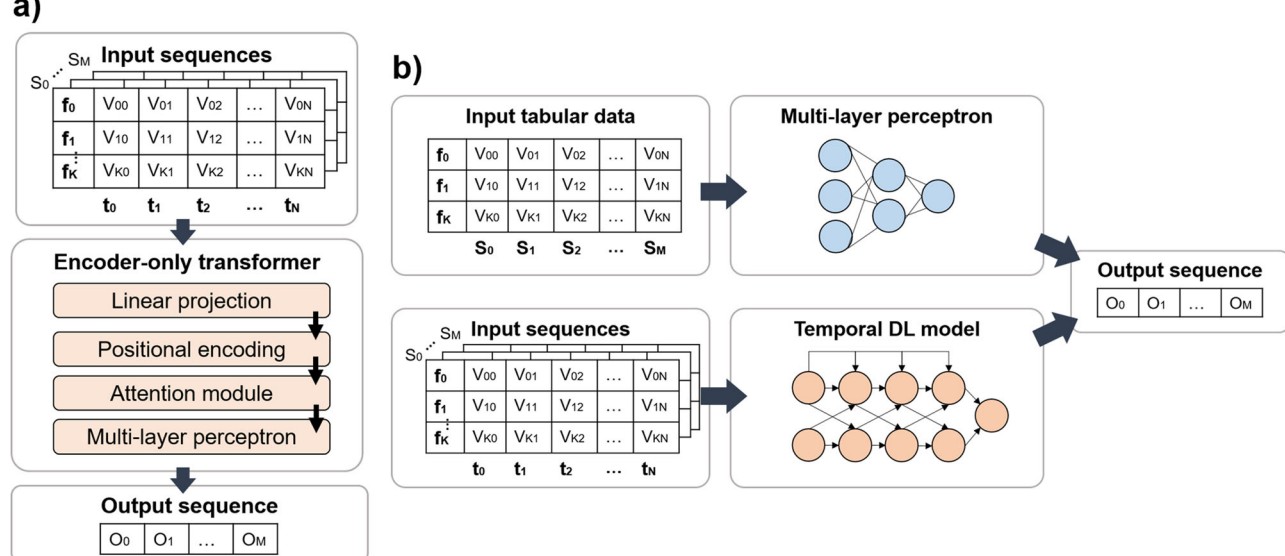

**Fig. 2 | Transformer - as well as combined model architecture.** Adapted encoder-only transformer architecture is shown in **a**. Input sequences are fed into a layer stack including a linear projection extending inputs to a learnable input space, positional encoding keeping index information, a multi-head attention module, and a MLP. The architecture outputs one scalar prediction per surgery. Panel **b** shows the architecture for combined models comprising a MLP and a temporal deep learning (DL) model (LSTM or transformer). Separate predictions are combined by a linear layer to form one output per surgery.

to a column vector $[BP_1, BP_2, \ldots, BP_{10}]^\top$. The resulting tabular data were labeled with TAB_T. Time-static data were always fed without aggregation into MLPs. We also trained models on combined time-static and time-dynamic data. In this case, static features were assigned constant values across time steps. Before feeding data into our models, we standardized them through z-transformation utilizing statistics calculated on the training data[52].

## Model architectures

We trained MLP[53], LSTM[34], TRAN[37], and combined models. Models were either fed with sequential, tabular, or both types of data. Supplementary Table 2 summarizes the model variants with corresponding data inputs.

MLPs consist of multiple layers containing neurons, also called perceptrons[33], that are connected to all neurons that reside on the subsequent layer. Neurons are equipped with an activation function that maps the input with a learnable weight and a constant (bias) to an output[54]. All neurons form collectively a fully connected feed-forward neural network containing input-, hidden-, and output layers. Cybenko et al. have shown that MLPs are suitable for describing complex non-linear functions[55]. MLPs learn dependencies from tabular data and can not directly learn temporal patterns in parameter sequences[33].

Instead of sequentially learning temporal information, the attention mechanism processes sequences all in once[56]. The so-called attention weights can be used to learn the similarity between one element to all other elements within one sequence (self-attention). A TRAN architecture makes use of the self-attention mechanism to encode input sequences into a latent space[37]. Different parts of input sequences are learned by a parameterized multi-head attention module running the attention mechanism in parallel and concatenating results. For generating output sequences, the latent space is reconstructed by a TRAN using the so called cross-attention (decoder) for comparing model in- and outputs[57]. Attention networks are usually used in the context of natural language processing (NLP) tasks where texts (also words or characters) should be generated on the basis of other texts (seq2seq setting)[37,58].

We provide an explanation of mathematical concepts behind LSTM and TRAN in the corresponding section Supplementary Methods 2 (also see Supplementary Figs. 3 and 4). While we used standard MLP and LSTM modules[59], we propose an adapted TRAN model architecture based on work

by Vaswani et al.[37] that we named transformer-based POD predictor (TRAPOD).

Panel a in Fig. 2 depicts the model architecture for TRAPOD following an encoder-only approach following Vaswani et al.[37,57]. As we here consider the task of binary target variable prediction (classification), the TRAPOD architecture does not contain a decoder module that would be required for constructing output sequences in a seq2seq setting. Let $F$ be our selected features with $F = \{f_0, \cdots, f_k\}$, $S$ be the included surgeries of our cohort with $S = \{s_0, \cdots, s_m\}$, and $T$ be the time steps per sequence with $T = \{t_0, \cdots, t_n\}$. The model input for one surgery was represented by a matrix $M = [v_{00}, v_{01}, \cdots, v_{kn}] \in \mathbb{R}^{k \times n}$. The combined data of $m$ surgeries serving as input to the model was summarized in a tensor $I \in \mathbb{R}^{k \times n \times m}$ (see Fig. 2).

The input $I$ was embedded into a higher-dimensional space via a trainable linear projection. In the original version of a TRAN, this projection is replaced by pre-trained token embeddings[37,41]. Afterwards, an encoding layer[60] added positional information that would have been lost in the attention module[37]. After the positional encoding, a multi-head attention module calculated attention weights encoding temporal dynamics. In contrast to the architecture by Vaswani et al., a fully-connected MLP was stacked on top of the encoder instead of a cross-attention block connecting to a decoder. Thus, the latent space resulting from the encoder could be processed to one output prediction $O_m$ per surgery. Errors for outputs were back-propagated through the TRAN architecture for model training.

We trained combined model architectures comprised of one MLP model and one temporal DL (LSTM or our proposed TRAN network) acting on time-static as well as time-dynamic data, respectively. Architectures are outlines in panel b as part of Fig. 2. Temporal DL models were fed with the previously defined data $I$, while MLPs processed an additional matrix $A = [v_{00}, v_{01}, \cdots, v_{kn}] \in \mathbb{R}^{k \times m}$ holding $k$ static features for $m$ surgeries. Each model predicted one output and the two outputs were combined with a linear layer into one prediction per surgery (see Fig. 2).

We describe in detail how we determined model hyperparameters in the next section. Supplementary Table 3 shows our final model architecture including the number of selected neurons and layers per model variant. We outline the MLP configuration as neurons per layer that we kept constant to reduce complexity while hyperparameter optimization. We determined the number of hidden neurons within one LSTM module and the number of modules that were stacked on top of each other (layer). The tailored

attention network TRAPOD was optimized for the number of encoder layers as well as the neurons and layers that we used in the MLP on top of the attention module. The linear projection's dimension was 128 for the combined TRAN architecture (MLP_TRAN_SEQTAB) whereas the stand-alone TRAN was optimized with the a linear dimension of 32 (TRAN_SEQ). We kept a single attention-head that was used in the multi-head attention mechanism for all TRAN models as a result from hyper-parameter optimization (see Supplementary Methods 2).

Our final best performing TRAPOD configuration that based on sequential data (TRAN_SEQ) consisted of a total of 16,340 parameters comprising 7648 parameters for the multi-head attention module, 8577 parameters for the linear projection and positional encoding, and 155 for the MLP. Our proposed architecture can be implemented by other teams that deal with clinical prediction tasks such as ours. We compared our models with openly accessible clinical models (see Supplementary Data 5). Additionally, we trained GRU-D[43] as a baseline.

### Model training, tuning, and validation

To assess the performance of our models, we utilized the area under the receiver operating characteristic (AUROC) curve[61] and the area under the precision-recall curve (AUPRC)[62]. Both metrics were suitable for evaluating classification models. The ROC curve is a representation of the true positive rate (sensitivity) against the false positive rate (1-specificity) for different model thresholds. AUROC scores of 0.5 indicate a chance-level prediction whereas a value of 1 presents a perfect prediction. The AUPRC evaluates predictions based on varying precision and recall values where chance-level prediction is determined by the prevalence (fraction of positive cases). Hence, the AUPRC is more meaningful for imbalanced problems such as ours[32].

Patients were randomly divided into train/test sets based on patient identifiers (pat_id). Thus, all surgeries of a patient were fully contained either in the training or test set, preventing estimation bias due to information leakage between train and test sets. The train and test set comprised 48,348 and 12,088 patients, respectively. Supplementary Data 4 outlines descriptive statistics for both sets.

Hyperparameter search was conducted via a $3 \times 3$ nested cross-validation (CV) process that effectively prevents over-fitting[63]. Training samples were split into three equally sized subsets, or folds, where one (33%) was used as validation and two (66%) were used as training sets at a time. In nested CV, this process is repeated within the initial CV training subset (outer folds) to generate three subsets again (inner folds). The best hyper-parameter configuration is then searched for in the inner CV via a hyper-parameter search approach that is explained below, and validated on outer folds. Supplementary Fig. 2 depicts validation results from outer CV folds per model variant. In the main manuscript, we report results as mean and 95% confidence intervals as estimated on 1000 bootstrapped test sets. Bootstrapping, as random sampling of test data with replacement, was used to test model robustness[64].

Within the inner CV, we applied the grid search[65] as well as hyperband search[66] algorithms. A grid search evaluates all possible combinations of hyperparameters. In contrast, Hyperband frames the parameter search as a multi-armed bandit problem[66,67]. The algorithm can be seen as a very advanced random search approach eliminating less promising parameter sets while optimizing run time. Due to class imbalance, we selected model configurations leading to the best outer validation AUPRC scores.

To address class imbalance while training, we used either a weighted binary cross entropy (CE)[68] or a focal loss (FL)[69] function. FL, which was proposed in the context of object detection, introduces the parameter $\gamma$ focusing on hard-to-classify examples from the minority class, extending CE. Denoting $\gamma$ as the focusing parameter, $p_t$ as the output probability and $\alpha_t$ as the weighting parameter at $t$, FL[69] is defined as

$$FL(p_t) = \alpha_t (1 - p_t)^{\gamma} \log(p_t). \quad (2)$$

We included the type of loss function (CE or FL), the selection of a sampling interval (3 or 5 min), the learning rate ($1e - 4$ or $1e - 5$) and the

batch size (32 or 64) in a Grid Search process. Other parameters like the number of layers and neurons as well as stacked modules were determined by Hyperband Search. The number of epochs were evaluated by early stopping that is interrupting the training process if the validation loss does not increase after 10 training epochs (patience)[70]. We always used a logistic sigmoid activation function in our MLP models, as it consistently achieved stable results during training[54].

In the Supplementary Table 4, we provide details on the hyperpara-meter search spaces. Supplementary Data 6 lists final model hyperpara-meters and trainable dimensions found during the nested CV. We also report training performances (see Supplementary Methods 3). Validation did not deviate strongly from training performance suggesting that there was no pronounced problem of over-fitting (see Supplementary Fig. 5). All our results are reported in accordance with the transparent reporting of a multivariable prediction model for individual prognosis or diagnosis (TRIPOD)[71] guidelines (see Supplementary Data 7).

### Statistical testing

Associations between univariate features and POD were assessed using Spearman's rank correlation coefficient[72] on training data. The method can be seen as applying Pearsons' correlation coefficient to ranks of data increasing robustness against outliers assuming a monotonic increase or decrease. A coefficient near $-1$ or $1$ indicates a very strong correlation, a coefficient of 0 indicates no correlation. We report absolute values of coefficients in our study due to their simple interpretation, and indicate the effect direction separately where it is of interest. A t-statistic assessed the statistical significance of Spearman correlations against the null hypothesis of zero correlation[73].

In addition to calculating correlations, we also fitted feature-wise mixed-linear effects models (MLEMs)[74] with POD as independent and each individual feature as the dependent variable, taking additionally other factors into account. Specifically, MLEMs offer the possibility to model data with hierarchical structure by properly accounting for dependencies between samples induced by grouping. In our data, such dependencies arise, for example, when multiple surgeries are observed for one patient. By including patient identifiers as a predictor, the model jointly fits a point-specific offset, thus correcting for this hierarchical structure[75].

We define our MLEM models with the following formula: $feature \sim C(target) + time + C(target) \times time + 1|pat\_id/op\_id$. Feature values or missingness rates (feature) are defined as the dependent variable. For fixed effects as independent variables, we configured our binary POD variable (target) (1 = POD, 0 = no POD) and the time index (time) (1–3) for three non-overlapping 30 min intraoperative windows. Additionally, we included interactions between time and POD ($C(target) \times time$) as fixed effect. As mentioned above, we corrected for patient-surgery hierarchies by introducing nested-random effects (1|pat_id/op_id).

We report coefficients for fixed effects and corresponding p-values. P-values were calculated using t-statistics for the null hypothesis of zero coefficients. To make results comparable, we transformed input and output data (from dependent and independent variables) with min-max normal-ization so that all values ranged between 0 and 1[52].

All p-values were corrected with the false discovery rate (FDR)[76] method for multiple testing, applying an alpha level of 0.05. For the correlation analysis, we applied the correction separately for each distinct intraoperative 30 min window. However, our cohort was very large increasing the chance of finding statistically significant results even for small effects.

To compare AUROC and AUPRC scores obtained from different models with each other, we used a corrected paired Student's $t$ test[73]. The unmodified Student's $t$ test assumes independent sampling from two populations. This assumption is violated by implementing bootstrapping due to the high dependency between sampled test sets. Estimator variances would be underestimated leading to type I errors. Nadeau et al. proposed a modified version of Student's $t$ test correcting variance estimations based not only on the test, but also on the training samples[73].

We wanted to compare vectors of performance metrics $P_0 = [p_{00}, \cdots, p_{0n}]$ and $P_1 = [p_{10}, \cdots, p_{1n}]$ for models M0 and M1 retrieved from $n$ bootstrapped test sets. The absolute differences between these two were calculated as $D = |P_0 - P_1|$ where the operator ( $-$ ) donates an element-wise subtraction. The variance $\sigma^2$ for these distances was scaled by the number of training ($n_{train}$) and testing ($n_{test}$) samples as

$$\sigma^2_{mod} = \sigma^2(D) = \left(\frac{1}{n} + \frac{n_{train}}{n_{test}}\right). \quad (3)$$

From this modified variance, which was proposed by Nadeau et al.[73], the t-statistic was derived as

$$t = \frac{\overline{D}}{\sigma^2_{mod}} \quad (4)$$

with $\overline{D}$ denoting the arithmetic mean of distances. Although this approach has also been suggested and evaluated previously by authors[77,78], we did not base our model comparison purely on this modified test statistic. Results are reported with confidence intervals, which are equally adequate to compare model performances[79].

### Statistics and reproducibility
We described our statistics comprehensively and provide executable Python code via our code repository[80]. Methods should be applied to the training dataset exclusively avoiding validation biases. Alpha levels for assessing statistical significance are included in Supplementary Data 8.

### Reporting summary
Further information on research design is available in the Nature Portfolio Reporting Summary linked to this article.

## Results
### Cohort characteristics
The study cohort consisted of 60,436 patients undergoing a total of 72,100 surgeries within 68,983 distinct hospital stays with a POD prevalence of 9.38%, 8.27%, and 8.47%, respectively (see Supplementary Fig. 6 and Supplementary Results 1). The presence of POD was defined as at least one Nu-DESC ≥ 1 measured in the recovery room (see Methods). On average, the Nu-DESC assessment was performed 1.2 ± 0.5 (mean ± std) times, where the first score was assessed 37.39 ± 26.83 min after the extubation for the whole patient cohort. Patients experiencing POD had prolonged stays in the hospital (11.45 ± 7.63 vs. 8.31 ± 5.78 days) and in the recovery room (2.51 ± 7.63 vs. 2.03 ± 2.63 h). They also underwent longer anaesthesia (4.65 ± 1.36 vs. 2.99 ± 1.24 h). While the mean age was higher for POD positive patients (60.28 ± 18.88 vs. 52.93 ± 18.22 years), there was no pronounced difference in the gender distribution (51.33 vs. 53.60% female). Patients experiencing POD seemed to be more frail with a weaker physical preoperative status (see Supplementary Table 5).

### Univariate testing
To investigate the relation of single perioperative features with POD, we conducted univariate analyses. Firstly, we calculated Spearman's rank correlation coefficients[72] ($r_s$) between mean feature values in different analysis windows and POD. The same analysis was carried out for feature missingness rates, defined as the percentage of missing measurements per feature, instead of feature means. Correlation coefficients were calculated per feature for different shifted non-overlapping intraoperative time windows covering 30 min intervals, and for the entire surgery duration. Windows were either shifted forward relative to the start of the surgery (T_begin) or backwards to the surgery's end (T_end) (see Supplementary Data 8). We describe results for the top 20 features whose mean values are most strongly correlated with POD during the entire surgery (see Fig. 3). We included features that potentially changed during the surgery (time-dynamic) and calculated means per analysis window. Statistical results were corrected for

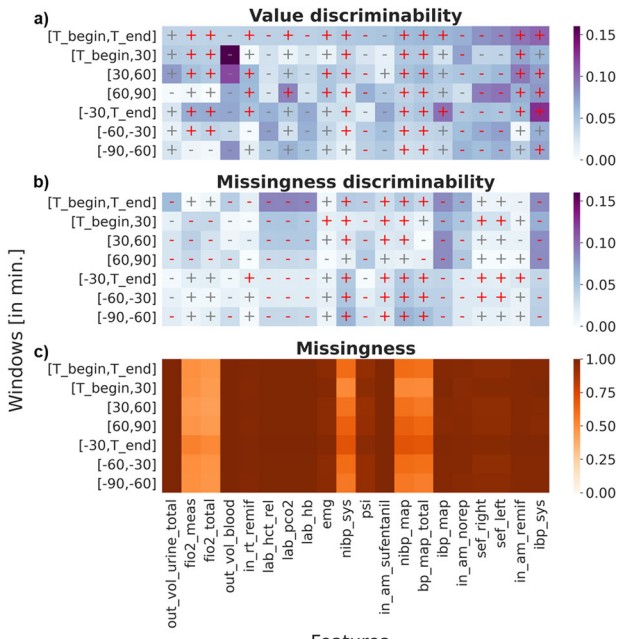

Fig. 3 | Spearman's rank correlations. POD discriminability of mean feature values and feature missingness (columns) in different observation time windows (rows). Besides the full intraoperative phase indicated by [T_begin, T_end], six further non-overlapping 30-min intervals were defined relatively to either T_begin or T_end. Discriminability was measured for each feature as the absolute value of Spearman's rank correlation between feature and POD occurence (shown in blue colors). The panel **a** displays correlations for mean aggregated feature values, while the panel **b** shows analogous correlations for missingness rates per feature and time window. The effect direction is indicated as positive (+) or negative (−). The panel **c** depicts the actual fraction of missing values (brown colors). Red color indicates significant results at an FDR corrected alpha level of 0.05. Results are shown for 20 features exhibiting strongest correlations with POD during the full intraoperative phase. $N = 48,348$ patients in the training set.

multiple testing by false discovery rate (FDR)[76] per analysis window (alpha level = 0.05).

Several vital signs and respiratory parameters, such as non-invasive mean blood pressure (nibp_map) and measured fraction of inspired oxygen (fio2_meas), were abundantly available during intraoperative time windows. Others, such as systolic invasive blood pressure (ibp_sys), hemoglobin in blood (lab_hb), and blood output (out_vol_blood), were rarely acquired (see panel **c** in Fig. 3).

Of all investigated features, ibp_sys was most strongly correlated with POD during the intraoperative phase (window [T_begin, T_end]: $r_s = 0.102$, p−value $= 2.38e^{-13}$), exhibiting a positive correlation that was most pronounced during the last 30 intraoperative minutes (window [-30, T_end]: $r_s = 0.126$, p−value $= 3.36e^{-11}$). The absence (missingness) of ibp_sys measurements was negatively correlated with POD across all time windows and strongest for the entire surgery (window [T_begin, T_end]: $r_s = 0.085$, p−value $= 7.18e^{-19}$). Hence, high ibp_sys values as well as the actual presence of invasively measured values were found to be positively associated with POD (see a and b in Fig. 3).

Additional features strongly correlated with POD within the complete intraoperative phase ([T_begin, T_end]) were: the amount of the opioid remifentanil given (in_am_remif) ($r_s = 0.096$, p−value $= 4.69e^{-32}$), the spectral edge frequencies (sef_left and sef_right) ($r_s = -0.095/ -0.093$, p−value $= 1.33e^{-18}/7.58e^{-18}$), and the given amount of norepinephrine (in_am_norep) ($r_s = -0.075$, p−value $= 4.77e^{-62}$) (see top graph in Fig. 3). Spectral edge frequencies as well as the so-called patient state index (PSI), which are used to estimate sedation depth[81], were derived from 4-channel electroencephalography (EEG). While blood outputs (out_vol_blood) recorded within the first 30 min appeared to show high discriminability, the

## Coefficients from Mixed Linear Effect Models

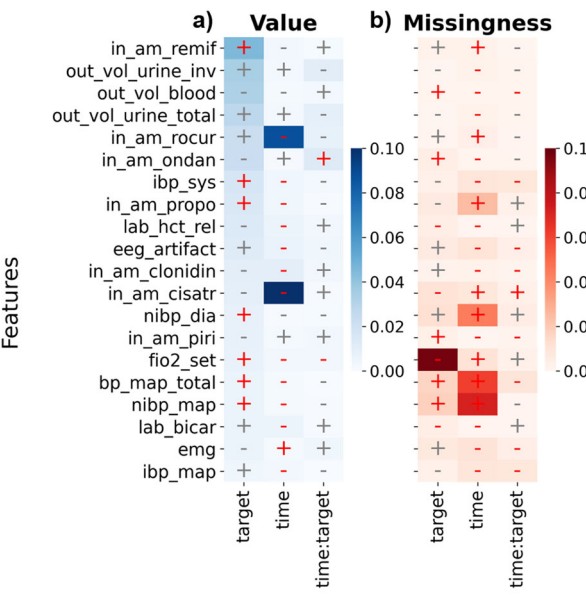

**Fig. 4 | Mixed linear effects.** Coefficients of mixed linear effect models (MLEMs) linking means of time-dynamic features within analysis windows to POD presence (target), analysis interval (time), and time x target interactions (time:target), modeled as fixed effects. A correction for the relationship of patients having multiple surgeries was introduced by nested random effects (not shown). Time encodes three consecutive 30-min non-overlapping interoperative analysis windows relative to the beginning of the surgery (T_begin). Model coefficients for fixed effects (columns) are shown per feature (rows). Models were fitted on normalized feature values (left/blue), shown in panel **a**, or corresponding missingness rates (right/orange), displayed in panel **b**. Effect directions are indicated with +/− signs drawn in red for effects that are statistically significant according at an FDR corrected alpha level of 0.05. Results are shown for 20 features exhibiting largest absolute coefficients for POD onset (target). N = 48,348 patients in the training set.

sample sizes were insufficient to ascertain statistical significance, resulting in elevated error margins (window [T_begin, 30]: $r_s = -0.179$, $p$−value = 0.316).

The absence of certain laboratory values, such as hemoglobin in blood (lab_hb) (window [T_begin, T_end]: $r_s = -0.086$, $p$−value = $7.65e^{-81}$), was negatively correlated with POD. Laboratory values showed similar patterns in missing discriminability over analysis windows (see Fig. 3). Early analysis windows were more discriminative than later ones. The first intraoperative 30 min ([T_begin, 30]) yielded the highest mean absolute Spearman coefficient of 0.307 in contrast to other windows.

We calculated correlation coefficients for time-static features being constant during the perioperative phase (see Supplementary Table 6). Results highlight the presence of dementia (curr_dem), high age (age) and prolonged planned surgery length (praem_oplen) as variables strongly associated with POD.

In addition to analyzing Spearman correlations, we implemented linear mixed effects models (MLEMs)[74]. Two MLEMs were fitted for each feature using either the (min-max normalized) mean feature value or the feature missingness rate within analysis windows as the dependent variable. In contrast to previously described univariate models, MLEMs were controlled for the hierarchical relation of patient and surgery by including the respective identifiers as nested random effects. Hence, we could increase statistical power by removing variance. The POD outcome (target) and the analysis time window (time) were included as fixed effects. The factor time encoded one of three consecutive 30 min non-overlapping windows relative to the surgery begin and was included to assess statistical differences between windows. We present model coefficients related to fixed effects for

the 20 features with largest absolute coefficients for factor target as estimated via separate MLEMs (see Fig. 4). Statistical results were again FDR corrected (alpha level = 0.05) (also see Supplementary Fig. 7).

The highest target coefficient ($c_{target}$) could be observed for the given amount of remifentanil (in_am_remif) ($c_{target}$ = 0.044, p-value = $1.11e^{-8}$). High levels of invasive systolic blood pressure (ibp_sys) were, similar to previous findings, positively associated with POD ($c_{target}$ = 0.016, $p$−value = $5.25e^{-5}$). The time coefficient ($c_{time}$) was negative ($c_{time} = -0.004$, $p$−value = $4.06e^{-9}$), indicating a decrease of ibp_sys levels over intraoperative time. Similar effect directions to ibp_sys could be observed for the amount of the narcotic propofol (in_am_propo, $c_{target}/c_{time}$ = 0.014/ −0.004, $p$−values = $6.53e^{-6}/1.50e^{-14}$), which was not identified via the Spearman correlation analysis. As another difference to the Spearman analysis, the set fraction of inspired oxygen (fio2_set) during mechanical ventilation was positively associated with POD ($c_{target}$ = 0.007, $p$−value = $1.03e^{-23}$), decreased over time ($c_{time} = -0.003$, $p$−value = $1.84e^{-207}$), and was interacting negatively for target and time ($c_{time:target} = -0.004$, $p$−value = $7.71e^{-27}$). Post-hoc interpretation of this interaction shows that the initial positive effect of fio2_set on POD discriminability flips for later time windows. Specifically, the effect of POD decreases by 0.004 units for one additional unit of time.

Just as actual feature values, missingness rates were strongly positively correlated with POD for fio2_set ($c_{target} = -0.097$, $p$-value = $3.84e^{-85}$) and increased over intraoperative time especially for non-invasive mean blood pressure (nibp_map) ($c_{time} = -0.071$, $p$−value = $4.78e^{-10}$). Similar to previous univariate results, the actual blood outputs (out_vol_blood) were not highlighted by the MLEM analysis, but the absence of recordings had a positive association to the POD target ($c_{target}$ = 0.001, $p$−value = $1.32e^{-7}$), and decreased over time ($c_{time} = -0.002$, $p$−value = $1.919e^{-149}$). As this holds for out_vol_blood, documented urine outputs were more present in later phases of the surgery, e.g., invasive outputs (out_vol_urine_inv) ($c_{time} = -0.002$, $p$−value = $4.081e^{-123}$). Time x target interactions were more often statistical significant for missingness rates than for actual values (9 vs. 2 times).

Averaging across all features, we observed stronger absolute temporal effects (mean $|c_{time}|$ = 0.0132) compared to target effects (mean $|c_{target}|$ = 0.0128). Mean absolute time coefficients were greater for missing rates (mean $|c_{time}|$ = 0.0137) than for feature values (mean $|c_{time}|$ = 0.0127).

### Machine learning models

We trained and applied multiple ML model variants comprising stand-alone long LSTM, TRAN, MLP architectures as well as combined models for POD predictions. While TRAN and LSTM modules ingested sequential (SEQ) data, MLPs were trained with tabular (TAB) data aggregating temporal changes in clinical signals. MLPs used either time series features like wavelet coefficients (MLP_TAB_F), summary statistics like percentiles (MLP_TAB_P), or vectorized (transposed) time series (MLP_TAB_T) (see Methods).

We obtained performance metrics on the test set across model variants ingesting the first intraoperative 30 min ([T_begin, 30]) (see Table 2 and Fig. 5). Here, we selected this observation window to compare model variants due to previously described univariate results and due to the high clinical relevance of an early risk assessment[2]. The overall best performing model implemented a stand-alone TRAN architecture using SEQ data (TRAN_SEQ). The model achieved a mean AUROC of 0.774 (95%-CI [0.711, 0.787]), a mean AUPRC of 0.330 (95%-CI [0.328, 0.340]), and a mean precision of 0.161 (95%-CI [0.156, 0.169]) at 0.8 recall on a bootstrapped test set. The TRAN_SEQ model also reached the highest sum of sensitivity and specificity with mean values of 0.711 (95%-CI [0.704, 0.718]) and 0.711 (95%-CI [0.709, 0.718]) respectively (see Table 2 and Supplementary Data 9). We observed pronounced differences in performances indicated by non-overlapping confidence intervals (CI) between TRAN_SEQ and all other models.

We evaluated the receiver operating characteristic and precision-recall curves from which AUROC and AUPRC metrics are derived (see Fig. 5).

Curves are almost non-overlapping for TRAN_SEQ with all other variants over varying prediction thresholds for both metrics. Differences in AUPRC were less pronounced between MLP based variants ingesting TAB data and between LSTM and TRAN (excluding TRAN_SEQ) models. These clusters were not visually apparent for AUROC, which, unlike AUPRC, is insensitive to the actual POD prevalence.

We analyzed distributions of AUROC and AUPRC scores retrieved from the bootstrapped test set across model variants (see Fig. 6). Results are also shown for statistical tests comparing mean performances between two models while controlling for dependencies between bootstrap samples. Performance of the TRAN_SEQ model was significantly higher compared to all other models. The MLP_TAB_P variant that aggregated temporal dynamics with summary statistics outperformed the MLP_TAB_F version that ingested more complex time series features. Temporal DL models (TRAN, LSTM) generally outperformed MLP models. Combined models and the stand-alone LSTM variant seemed to perform equally good but not as good as the stand-alone TRAN_SEQ model (see Fig. 6). Due to its superior performance as measured, we designate our best trained TRAN_SEQ model as the transformer-based POD predictor (TRAPOD).

We further trained models on data from different observation windows of varying lengths and positions within the intraoperative phase (see Supplementary Results 2). We did not clearly identify an observation window with superior POD prediction performance (see Supplementary

### Table 2 | Model performances

| Model | AUROC | AUPRC |
|---|---|---|
| MLP_LSTM_SEQTAB | 0.743 [0.741, 0.744] | 0.259 [0.256, 0.261] |
| MLP_TRAN_SEQTAB | 0.740 [0.738, 0.741] | 0.254 [0.251, 0.256] |
| LSTM_SEQ | 0.749 [0.748, 0.751] | 0.263 [0.260, 0.266] |
| TRAN_SEQ | 0.774 [0.772, 0.787] | 0.330 [0.328, 0.340] |
| MLP_TAB_F | 0.711 [0.709, 0.712] | 0.205 [0.204, 0.207] |
| MLP_TAB_P | 0.717 [0.715, 0.718] | 0.221 [0.219, 0.223] |
| MLP_TAB_T | 0.707 [0.705, 0.709] | 0.209 [0.207, 0.211] |

AUROC and AUPRC metrics per model variant (Model) calculated on a 1000× bootstrapped test set. All models were trained and applied on time series retrieved from the first intraoperative 30 min. ([T_begin, 30]) with a three minutes sampling interval. Results are reported as mean [95%-CI]. Multilayer perceptrons (MLP), transformers (TRAN), long short-term memory (LSTM) networks, and combinations of these were included in the comparison. Each model processes either sequential (SEQ) data, aggregated tabular (TAB) data, or both. Time series data were aggregated in different ways to yield tabular data.

Figs. 8 and 9, Supplementary Table 7). However, we observed overall higher AUPRC scores during the first 30 min of a surgery ([T_begin, 30]). Hence, we conducted further analyses with this window.

We compared our proposed ML approaches with existing prognostic baseline POD prediction models accompanied by published model specifications. Concretely, we implemented six openly accessible LR approaches (by Wassenaar, Boogard, Kim, Zucchelli, Vreeswijk, and Xing et al.) with aggregated time series data, whenever used, from the observation window [T_begin, 30][21–26]. We applied each baseline model to the test set using published model coefficients. Additionally, model coefficients were re-estimated on training data (see Supplementary Results 3 and Supplementary Table 8). Among the baseline models, the retrained LR model of Kim et al.[23] achieved the highest AUROC with a mean of 0.637 (95%-CI [0.636, 0.637]). The published LR based on Zucchelli et al.[25] reached the highest mean AUPRC with 0.177 (95%-CI [0.177, 0.178]). Note that these performances are substantially inferior to the results obtained using non-linear DL models on temporal features, as presented above. This is also apparent from the corresponding ROC and PRC curves obtained for baselines, which are substantially lower than the ones from other model variants (see Supplementary Fig. 10).

In addition to simple open clinical prognostic models, we trained GRU-D on data from window [T_begin, 30] (see Supplementary Results 4). The model explicitly learns so-called decay rates describing more advanced missingness patterns than last observation carried forward (LOCF)[49] that is commonly used for imputing unobserved values in multivariable time series. GRU-D achieved a mean AUROC of 0.747 (95%-CI [0.731, 0.751]) and a mean AURPC of 0.282 (95%-CI [0.271, 0.293]) (see Supplementary Fig. 11). These performances line up between our best performing MLP (mean AUROC 0.717 95%-CI [0.715, 0.718], mean AUPRC 0.221 95%-CI [0.219, 0.223]) and TRAPOD (mean AUROC 0.774 95%-CI [0.772, 0.787], mean AUPRC 0.221 95%-CI [0.328, 0.340]), similar to combined model variants (see Table 2). Analyzing decay rates with high temporal signal for input variables highlighted features that were also identified by previous univariate tests (see Supplementary Fig. 12).

To provide insights into TRAPOD, we converted its attention weights into a matrix with dimensions features × time points (see panels a and b in Fig. 7). Attention weights (AWs) play a substantial role for the TRAN architecture with respect to learning temporal dependencies in sequential time series[37]. Weights are scaled through a softmax function so that they are non-negative and assume values between 0 and 1 (see Supplementary Methods 2). By visualizing AWs, we could identify temporal patterns relevant for the TRAN learning process (attention).

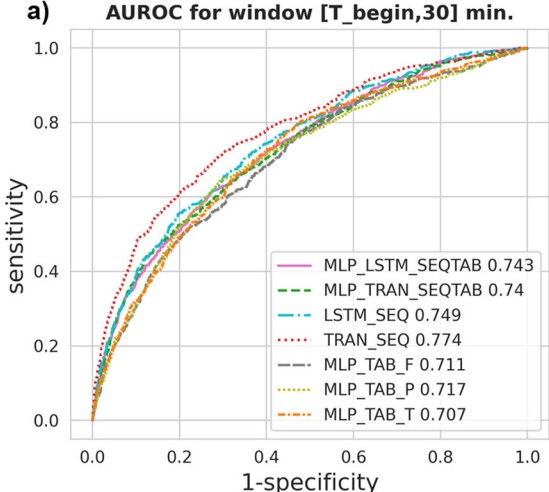
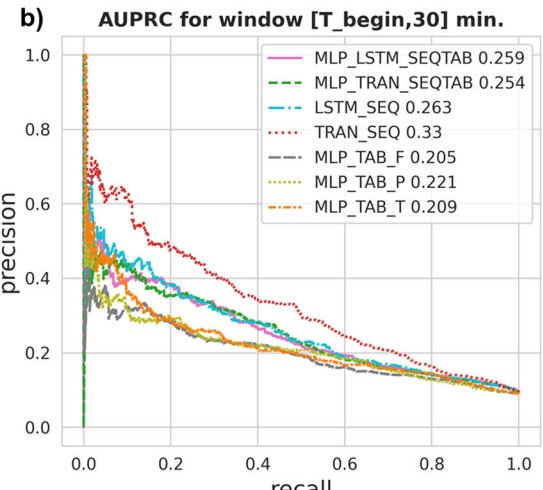

**Fig. 5 | Prediction threshold curves.** Test performance metrics across model variants applied to data from within the first intraoperative 30 min ([T_begin, 30]). AURO curves, panel **a**, and AUPR curves, panel **b**, are shown for each model. Multilayer perceptrons (MLP), transformers (TRAN), long short-term memory (LSTM) networks, and combinations of these were included in the comparison. Each model processes either sequential (SEQ) data, aggregated tabular (TAB) data, or both. Time series data were aggregated in different ways to yield tabular data.

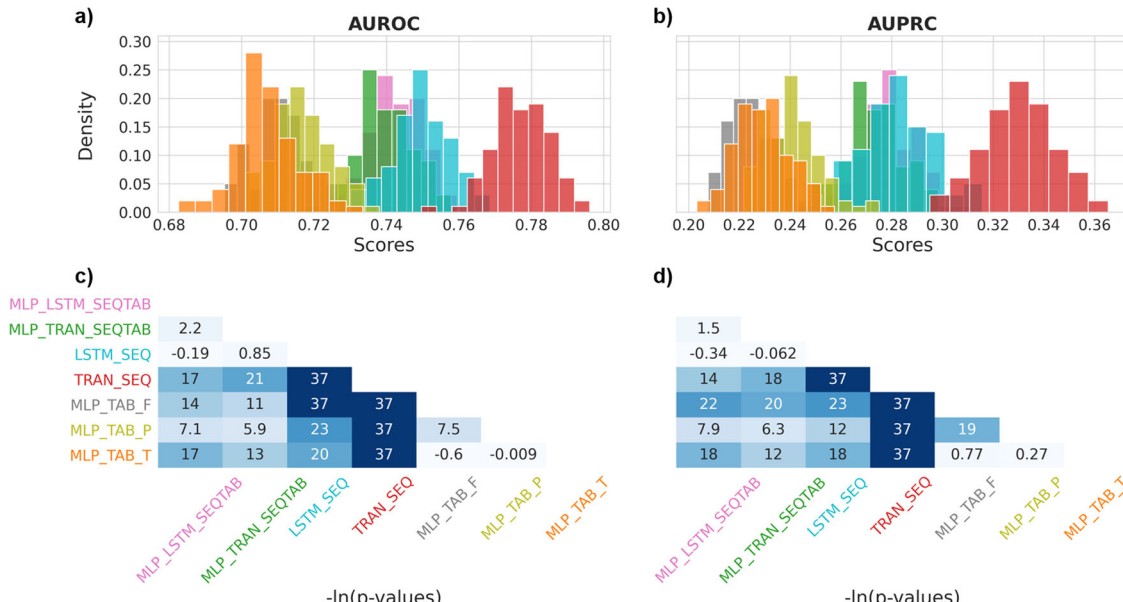

**Fig. 6 | Performance confidence.** Performances across model variants (colorized) applied to the first intraoperative 30 min [T_begin, 30] as calculated on a 1000× bootstrapped test set. Distribution of AUROC and AUPRC scores are shown in panels **a** and **b**. A corrected pairwise Student's *t* test was used to calculate *p*-values for the null hypothesis of equal mean performance of pairs of two models. The -ln(p-value) was retrieved for AUROC and AUPRC scores, shown in **c** and **d**. Multi-layer perceptrons (MLP), transformers (TRAN), long short-term memory (LSTM) networks, and combinations of these were included in the comparison. Each model processes either sequential (SEQ) data, aggregated tabular (TAB) data, or both. Time series data were aggregated in different ways to yield tabular data.

Similar to the previously described univariate feature importance assessment, the attention analysis identified invasive systolic blood pressure (ibp_sys) as relevant for the prediction of POD. The AWs for this feature attained a maximum of 0.197 at 15 min and a total sum of 0.794 over time. Levels of partial pressure of oxygen in blood (lab_po2), which were also highlighted by Spearman correlations, contributed to the TRAPOD's attention during the middle of the 30-min window (AW of 0.196 at 15 min). The given amount of the opioid remifentanil over time (in_rt_remif), which was also identified as important in the univariate analysis, was assigned to attention weights that peaked at 9 min (0.121) and 12 min (0.145) (see Fig. 7). In contrast to univariate results, the attention mechanism focused its weights on the recorded pulse (puls) (sum AWs = 0.568), the given dose of the antibiotic cefazolin (in_am_cefazol) (sum AWs = 0.569), as well as the infused volume of electrolytes (in_vol_elyts) (sum AWs = 0.490). The missingness rates of hemoglobin in blood (lab_hb) were received attention at 12 min and 15 min with AWs of 0.156 and 0.153, respectively. When averaging sums of AWs (aggregating time steps) over all features, feature values yielded a mean of 0.423 whereas feature missingness resulted in a mean of 0.419.

We averaged AWs across features per time index (see panel **c** in Fig. 7). Mean AWs were specifically elevated at 12 (mean 0.049), 15 (mean 0.052), and 21 min (mean 0.051). Summing displayed mean AWs for time indexes 3–15 yielded 0.172, for 18–30 the sum was at 0.159. These results imply that the attention mechanism puts more weight on the first half of the intraoperative 30 min window than on the second one (see Fig. 7).

**Scenario analysis**

Early detection of POD after the initial 30 min of the surgery potentially allows preventive clinical countermeasures, like transfers to specialized delirium wards, to be implemented. The analysis of prediction windows yielded a median of 2.26 h enabling these countermeasures (see Supplementary Fig. 13). Varying prediction windows lengths were caused by different Nu-DESC assessment points in time and varying surgery lengths.

Assuming 100 surgeries per day, we calculated confusion matrices for three different threshold configurations either A) maximizing sensitivity and specificity, B) maximizing precision and recall, or C) setting recall to 0.8 (see Fig. 8). In Scenario A, the model configuration led to a sensitivity and specificity both at 0.711. Scenario B achieved the highest recall out of all scenarios with 0.974 leading to 8 True Positives (TP) out of 9 Positives at the cost of 66 False Positives (FP). The number of FP could be decreased to 36 in Scenario C at 0.163 precision.

Our aim was to find a balance between overtreatment and patient safety. High rates of overtreatment (FP), as treating persons not suffering from POD, lead to increased short-term hospitalization costs[1]. Low patient safety (TP), as missing to treat patients who experience POD, can lead to poor outcomes such as dementia[4], and can incur long-term healthcare costs. In this respect, Scenario C was evaluated as the most favorable, leading to 7 out of 9 correctly identified POD positive cases and 55 out of 91 correctly identified negative ones.

**Discussion**

We assessed the discriminative value of temporal dynamics in time-dynamic features. Using univariable Spearman correlations on mean values per non-overlapping intraoperative time window. We identified blood pressure, given medications, and EEG features to be correlated with POD. Existing work has highlighted predisposing POD risk factors including high age, reduced physical activity, and preexisting dementia. These were confirmed in our univariate analyses[5,11,21,82]. Several research groups including Röhr et al.[83], Sun et al.[84], and Palanca et al.[81] have recently used EEG features for delirium prediction. Although EEG signals tend to be noisy showing artifacts[85], our models were also able to benefit from these signals[83,86] advocating further interoperative EEG monitoring.

We have also trained multivariable mixed-linear effect models (MLEMs) that correct for patient-surgery relations by introducing random effects while including time as a factor. Models could remove nuisance variance improving statistical power[87], hence leading to different results compared to Spearman correlations. Given opioids, narcotics, and collected body fluid outputs were identified by our multivariable analysis as discriminative with respect to POD. Analgetic opioids have also been discussed in delirium prevention guidelines[2] and by other authors[1,21,88,89], but the direction of their association with POD remains unclear[89].

In our setting, univariate results suggested a correlation between feature missingness rates and POD discriminability. The high availability of respiratory parameters and vital signs were in line with patient monitoring

## Attention weights

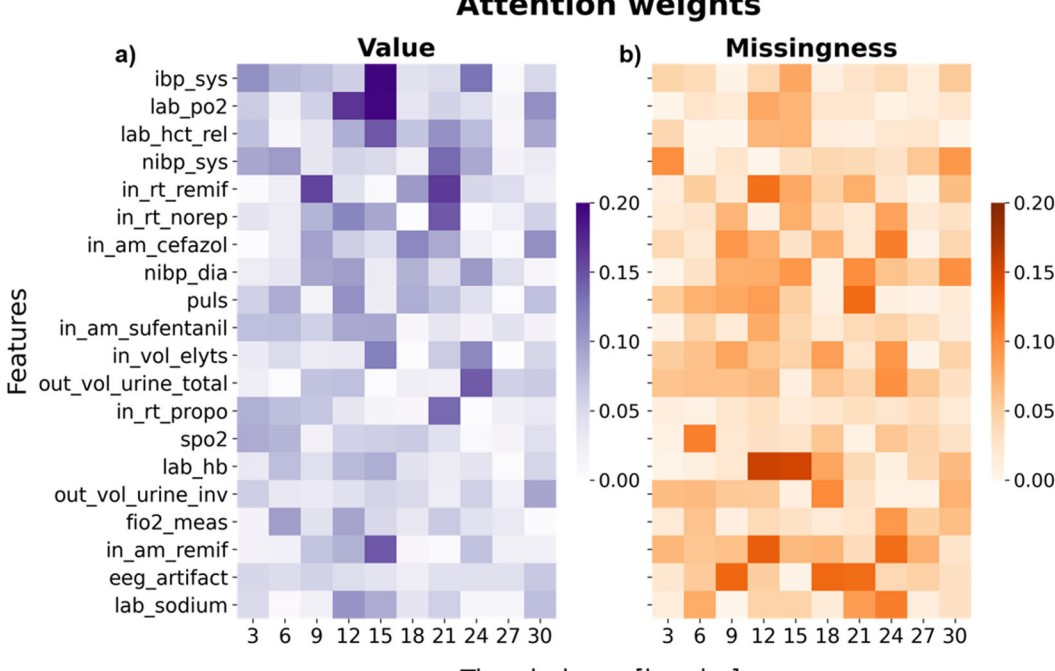

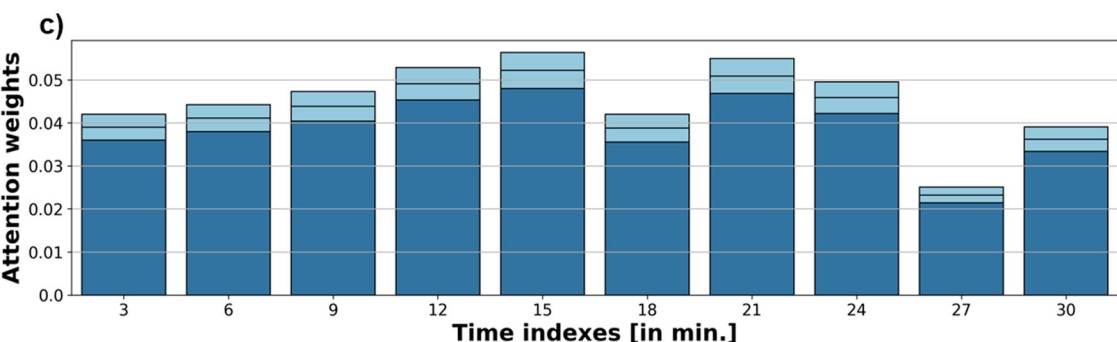

**Fig. 7 | Attention weights.** Attention weights trained with a transformer model ingesting sequential data (TRANS_SEQ) from the first intraoperative 30 min ([T_begin, 30]). Weights from the input layer were multiplied through a multi-head attention module to yield 10 time indexes (30 min at a 3-min sampling interval). Weights are displayed for time-dynamic features values in panel **a** or for the corresponding missingness indicator in panel **b** for 20 features whose feature values are characterized by largest summed attention scores over time. Aggregated attention weights across features per time index as mean [95%-CI] drawn as boxes on top of bars are shown in panel **c**. N = 48,348 patients in the training set.

procedures during surgeries[90]. The absence of invasive systolic blood pressure (ibp_sys) was positively correlated with POD. The presence of invasive ibp_sys measurements can be seen as a proxy for more serious diseases but also for the invasiveness or extent of the surgery[6]. We encoded missingness information via binary indicator variables used as inputs alongside the actual feature values when training models.

As expected, trained DL models, capable of directly processing sequential time series, outperformed MLP models working with aggregated tabular data. Contrary to anticipated findings, the MLP (MLP_TAB_P) that summarized temporal dynamics by simple statistics, like percentiles, outperformed the MLP model (MLP_TAB_F) that used more advanced temporal features like wavelet coefficients[51]. We observed high fractions of missing values for many time-dynamic features, which could have impeded the detection of rhythmic activity by more advanced methods. In contrast to Liu et al.[30], combined models fusing temporal deep-learning (LSTM or TRAN) and MLP models did not turn out to be superior to stand-alone architectures in our analysis. We also compared our proposed ML architectures to six published prognostic models. These baselines performed inferior to any of the proposed models.

In this study, the best performing attention-based network TRAPOD reached a mean precision of 16.1% at 0.8 recall. LSTM models by

Bhattacharyya et al. achieved a maximum of 14.4% precision at the same recall[18]. Liu et al. reported performances of 75.1% precision at 0.8 recall also using a LSTM architecture[30]. Both authors trained their model variants with intensive care unit (ICU) data instead of intraoperative time series and reported delirium prevalences of 11–20%[18,30]. Since we dealt with narrowly monitored surgical patients, we could assess the perioperative parameters at a sampling interval of 3 min, while ICU data used by Liu and Bhattacharyya[18,30] were sampled hourly. In contrast to Liu et al.[30], we used less features (896 vs. 238), facilitating the clinical applicability of our models. Furthermore, we make all models openly accessible[80]. Similar to Bhattacharyya et al., we analyzed the impact of different observation windows on our model performance[18]. We did not find distinct differences between windows with respect to POD discriminability, though. However, different data sources, clinical settings, and prevalences made it impossible to compare performances directly to previously published models.

We propose a simple TRAN encoder-only architecture TRAPOD for processing clinical multivariable time series (see Methods). TRAN models are complex models[91] and are therefore difficult to train. TRAPOD clearly outperformed other model variants archiving a mean AUROC of 0.774 and a mean AUPRC of 0.330. These results mark an improvement over the state of the art considering the very demanding prediction setting characterized

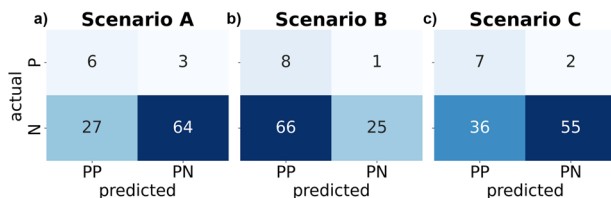

**Fig. 8 | Prediction scenarios.** Confusion matrix for three different scenarios showing the absolute number of actual negatives (N) and actual positives (P) in contrast to predicted positives (PP) and predicted negatives (PN) in POD prediction setting. Scenarios assume 100 surgeries with 9% prevalence leading to 91 AN and 9 AP. The prediction threshold is set as follows: maximizing sum of sensitivity and specificity in Scenario A, maximizing sum of recall and precision in Scenario B, 0.8 recall in Scenario C. Confusion matrices for scenarios are displayed in corresponding panels **a**–**c**.

by the low POD prevalence of 8.27% and a large heterogeneous cohort. Previous work mainly used TRAN architectures for generating representations, like learnable embeddings, that encode temporal dynamics in multivariable time series[40–42]. Other authors used more traditional techniques such as temporal association rules to learn temporal dynamics[92,93]. A systematic review by Xie et al. investigated DL temporal representations identifying data missingness as an important impediment[44]. We also observed a substantial number of missing values in our clinical data but successfully applied a TRAN encoding unobserved values with a binary missing indicator variable per time step[37].

The investigation of attention weights and their temporal patterns provided certain insight into the features combined by the model to derive at its POD prediction[94]. Specifically, the invasively measured systolic blood pressure and the application of remifentanil were highlighted by the transformer's attention weights during the middle of the first 30 intraoperative minutes. Our two univariate analyses (Spearman and MLEM) revealed that high levels of both variables over time were positively correlated with POD. The interpretation of attention weights alone without considering univariate importance could lead to false conclusions as it is not guaranteed that features with high AWs are at all statistically related to the prediction target. This limitation also affects other explainable artificial intelligence (XAI) methods like LIME and SHAP[95,96]. Additionally, it cannot be inferred that, in our extracted data, the associations of the identified features with POD are causal. In this regard, we could not differentiate between treatment indication and treatment effect (see Supplementary Discussion 1).

Our work has several limitations. First, we extracted data from only one clinical center with three different sites. Results would be more generalizable if applied across centers[97]. Our POD definition relied purely on the Nu-DESC assessment in the recovery room. Hence, we could not prevent the occurrence of possible documentation bias. However, taking Nu-DESC scores measured in the recovery room enabled the investigation of clearly defined prediction time windows that were seen as clinically valuable. In contrast to the Nu-DESC, that is required for recovery room discharge, the CAM-ICU can occur at any point in time on ICU. We understand that a single assessment score recorded in the recovery room serves as a snapshot and does not reflect POD's fluctuating symptoms over a longer time period. Multiple follow-up assessments may have yielded different outcomes.

Clinical information system (CIS) are not designed for retrospective clinical studies oftentimes resulting in distributed non-standardized data collection and storage[98]. We extracted and integrated 148 clinical variables (see Methods) that are available in front-end views of the CIS. To verify our extracted time series, we compared these with information displayed in our front-end views for 100 cases. Previous clinical studies have highlighted the importance of composite predictors like the Charlson Comorbidity Index (CCI)[99]. In our study, selected features relied on routinely collected data omitting these composite variables. Instead, we trained models with the history of International Classification of Diseases (ICD)[100] codes that can be freely combined by our models for addressing the POD prediction task. We

observed that trained models benefited from the inclusion of multiple inter-correlated time series. We did not apply advanced mathematical methods, like causal inference models[101], to correct for confounding. However, we believe that the inclusion of features that are not necessarily associated with our target variable, such as suppressor variables[102], potentially increased the predictive power of our models.

We have not validated our models on novel data extracted after 2020 so far. Since then, clinical practices, corresponding health record schemas, and patient characteristics might have changed due to the implementation of novel guidelines related to disruptive events such as a pandemic or the emergence of new disease variants[2,103]. We could also not validate our findings to openly available datasets due to the lack of perioperative data. The single-center design is a clear limitation but we hope to collaborate with other clinical centers for external validation. We acknowledge that additional model architectures, such as CNN-based models[42,104], could potentially be configured for predictions on clinical time series data. However, we believe that LSTM and TRAN models are well-suited to capture temporal dynamics for learning our POD prediction task. At this point, it is unclear whether the TRAPOD architecture generalizes to other clinical predictions tasks and settings. Future work will assess its efficacy for additional clinical predictions and validate it in multi-centric studies.

Our scenario analysis demonstrates the need for outbalancing patient safety and treatment costs. We suggest one possible model configuration that could guide clinical decision makers. However, especially over-treatment that arises for low prevalences, such as 8.27% here, could hinder implementation of TRAPOD in clinical practice. Prediction window lengths are considered clinically useful (see Supplementary Discussion 2).

We are currently developing an operational dashboard with data extracted after 2020 for a prospective POD risk analysis to explore potential challenges during the clinical translation process. Future prospective studies will directly assess the added value of TRAPOD for clinical decision support. Hereby, we hope that the clinical relevance of TRAPOD can be validated in practice facing obstacles like delayed - or bad data, low prevalence rates, or flawed documentation, that are observed in routinely collected health records[98]. Grigsby et al. extended the traditional TRAN design by introducing spatiotemporal attention layers[105]. Future work will take advantage of such recent advances of TRAN modeling for clinical prediction tasks[39,106,107]. To our knowledge, there has not been any published work applying a TRAN architecture to a perioperative prediction problem. Our models are openly available[80] and we are open for evaluation in other clinical centers.

In this retrospective study, we systematically investigated the role of temporal dynamics in intraoperative parameters for the highly clinically relevant POD prediction task. POD was measured in the recovery room after a surgery and defined with the Nu-DESC. We trained and applied MLP, LSTM, TRAN, and combined models. A TRAN architecture, TRAPOD, acting directly on the provided time series data without signal aggregation, outperformed all other model variants. We assessed the informativeness of individual features and temporal windows using univariate analyses, and compared the results to learned attention weights of the TRAN model. Additional considerations regarding patient safety and treatment costs need to be made before translating TRAPOD into clinical practice.

## Data availability
Data sharing is not possible due to German data privacy and protection regulations at our clinical institution. Summary statistics and statistical results for train - and test datasets can be downloaded via Dryad[108]. Chart related data are also included in Supplementary Data 4, 8, and 9. Direct data access for conducting similar analyses as presented can be requested via mail to the corresponding author.

## Code availability
Code and trained models can be found in our public GitHub repository[80] including usage instructions. Python code can be used with health data from other institutions. Further information can also be found in the Dryad repository[108].

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

## Acknowledgements
N.G. is funded by the German Academic Scholarship Foundation. M.S. is part of the Berlin Institute of Health at Charité - Universitätsmedizin Berlin, BIH Academy, Clinician Scientist Program. The authors acknowledge the Scientific Computing of the IT Division at the Charité - Universitätsmedizin Berlin for providing computational resources that have contributed to the research results reported in this paper URL: Charité - IT Division. This project has received funding from the European Research Council (ERC) under the European Union's Horizon 2020 research and innovation program (Grant agreement No. 758985). Results have been supported by the Metrology for Artificial Intelligence in Medicine (M4AIM) program of PTB that is funded by the German Federal Ministry for Economy and Climate Action (BMWK) in the frame of the QI-Digital Initiative.

## Author contributions
N.G. extracted, preprocessed, and cleansed the data. N.G. also trained and validated prediction models and applied statistical test methods. S.H., S.B., and N.G. conceptualized the methods design regarding ML models. F.B., C.S., and M.S. assisted with clinical input. K.R., S.B., and S.H. ensured the quality of statistical methods.

## Funding

## Competing interests
The authors declare the following competing interests: S.B. is a salaried employee at Pfizer Pharma GmbH and a visiting researcher at IMI. The majority of this work occured while Sebastian was at IMI exclusively.
