## [Transparent Peer Review file · Communications Medicine]

Applying a transformer architecture to intraoperative temporal dynamics improves the prediction of postoperative delirium

Corresponding Author: Mr Niklas Giesa

Version 0:

Reviewer comments:

Reviewer #1

(Remarks to the Author)

TRAPOD

This retrospective study in more than 60.000 patients is intended to investigate the temporal dynamics in intraoperative parameters that are associated with prediction of POD using a machine learning approach.

The proposed approach is interesting and addresses a clinically relevant topic.

Despite the clinical relevance and implication, the manuscript needs to be extensively revised for clarity. Furthermore there are some methodological limitations that should be appropriately acknowledged.

Major comments

The Authors report that the diagnosis of POD was based on NuDESC. It is important to report the actual length of postoperative monitoring and the daily checking. POD is a fluctuating symptom and accuracy in diagnosis is a key element to frame related risk factors.

The variables tested by the Authors are among those routinely monitored in the peri and intraoperative period but do not include some of those that are "proven" to relate with POD (check available literature).

Of specific interest, is the evidence on the relationship between arterial blood pressure and POD that several studies do not confirm as a POD risk factor. Is not clear if –according the Authors- the presented results should replace or complete the available evidence.

Inadequate screening for relevant variables (preoperative Comorbidity and cognitive function, etc) might include a bias that possibly prevent to confirm the presented association.

Is not clear if the Authors consider that the reported accuracy (AUC 0.7-0.8) is adequate to design specific clinical decision for the subgroup of patients at "higher risk" in a prospective study.

Minor comments

The text is presented as explanation of figures. Rather the Authors should frame the content to deliver to the readers and use the figures to clarify the information.

Sentences should –generally- not begin with a "Figure" rather should end (where appropriate) referring to a figure.

Some of the abbreviations are repeated in the text after the first use.

Page 2: long short-term memory (LSTM),

Page 6: long short-term memory (LSTM),

Similarly, when an abbreviation is presented , it should be consistently used thereafter

Page 2: Multiple studies used non-linear machine learning (ML)

Page 7: machine learning

Authors have the responsibility for checking the manuscript accuracy.

Reviewer #2

(Remarks to the Author)

This paper explores transformers to learn temporal information in dynamic features to predict post-operative delirium. The paper concludes that the developed model, TRAPOD, outperformed other models. The major weaknesses are that the paper's contribution lacks novelty and needs more baselines for comparison.

Strengthes:

- 1) It aims to solve a clinically meaningful problem with real patient data. The developed TRAPOD model, based on the transformer architecture, achieves notable improvements in performance against the baselines included.
- 2) It evaluates the model using temporal information from different time intervals prior to or during the intraoperative period for early detection. The statistical analysis is detailed and properly formulated, potentially providing insights into model explanation using attention weights.
- 3) Overall, the paper is well-structured and easy to follow.

Weaknesses:

- 1) The evaluation needs to include more experiments to compare TRAPOD with other models. For example, XGBoost has shown robust performance on EHR datasets in previous studies. Among deep learning models, CNN and other variations of recurrent neural network such as GRU-D should be included in the performance comparison.
- 2) The work does not have enough novelty to inspire future developments in the field. The utilization of a transformer encoder combined with MLP to learn dynamic and static data representations is a well-established approach to multi-modal clinical data, and the conclusion that it outperformed LSTM and MLP is not surprising. The data preprocessing including imputation is quite standard as well. While the authors claim novelty in its encoder-only structure, the contribution is not sufficient.

Version 1:

Reviewer comments:

Reviewer #1

(Remarks to the Author)

The Authors have addressed the points raised by reviewers.

Reviewer #2

(Remarks to the Author)

The proposed model demonstrated better performances compared to the newly added baseline experiments, addressing my concerns raised in previous reviews. I agree that the enhanced result section now is more robust and comprehensive. The authors also acknowledged model's limitation and emphasized on its superior performance tailored for the specific task of Delirium classification in a single site study.

Overall, the contribution of the work is now clearer, and the discussion of potential limitations is thorough enough to inform and inspire future research in this area.

Responses to referee's expertise

Referee expertise:

- > Referee #1: clinician, post-operative delirium
- > Referee #2: machine learning

Reviewers' comments:

Reviewer #1 (Remarks to the Author):

- > This retrospective study in more than 60.000 patients is intended to investigate the temporal dynamics in intraoperative parameters that are associated with prediction of POD using a machine learning approach.
- > The proposed approach is interesting and addresses a clinically relevant topic.
- > Despite the clinical relevance and implication, the manuscript needs to be extensively revised for clarity.
- > Furthermore there are some methodological limitations that should be appropriately acknowledged.

Major comments

COMMENT #1: The Authors report that the diagnosis of POD was based on NuDESC. It is important to report the actual length of postoperative monitoring and the daily checking. POD is a fluctuating symptom and accuracy in diagnosis is a key element to frame related risk factors.

RESPONSE #1: Thank you very much for your comment. We completely agree that POD is a fluctuating symptom and reported additional details on the postoperative monitoring and screening procedures.

Specifically, we added "On average, the Nu-DESC assessment was performed 1.2 ± 0.5 (mean \pm std) times, where the first score was assessed 37.39 ± 26.83 minutes after the extubation for the whole patient cohort. Patients experiencing POD had prolonged stays in the hospital (11.45 ± 7.63 vs. 8.31 ± 5.78 days) and in the recovery room (2.51 ± 7.63 vs. 2.03 ± 2.63 hours). They also underwent longer anesthesia (4.65 ± 1.36 vs. 2.99 ± 1.24 hours)." In Table A.1. in Appendix A, we report additional patient characteristics including perioperative durations.

Our target variable relied on the Nu-DESC measured inside the recovery room as our aim was to predict early POD onsets. Hereby, we wanted to help clinicians to identify vulnerable patients that might get transferred to a higher level of care [1]. Nu-DESC scores were assessed by trained nursing staff after the patient regained a certain level of consciousness. Daily follow-up checkups in ICUs were not considered so we focused on a heterogenous patient cohort while coherently defining our POD endpoint.

In the **Limitations** section, we explicitly highlight concerns that you have raised: "Our POD definition relied purely on the Nu-DESC assessment in the recovery room. Hence, we could not prevent the occurrence of possible documentation bias. However, taking Nu-DESC scores measured in the recovery room enabled the investigation of clearly defined prediction time windows that were seen as clinically valuable. (...) We understand that a single assessment score recorded in the recovery room serves as a snapshot and does not reflect POD's fluctuating symptoms over a longer time period. Multiple follow-up assessments may have yielded different outcomes."

[1] Spies, C., Piazena, H., Deja, M., Wernecke, K. D., Willemeit, T., Luetz, A., & ICU Design Working Group. (2024). Modification in ICU design may affect delirium and circadian melatonin: a proof of concept pilot study. *Critical Care Medicine*, 52(4), e182-e192.

COMMENT #2: The variables tested by the Authors are among those routinely monitored in the peri and intraoperative period but do not include some of those that are “proven” to relate with POD (check available literature).

RESPONSE #2: Thank you for expressing concerns about the selected variables. We extensively screened available literature. Composite scores, like the Charlson Comorbidity Index (CCI) [2] or dementia-related tests like the Mini-Cog [3] or the Mini-Mental-Status-Test (MMST) [4], were identified as strong predictors by previous clinical studies.

Our clinical institution has not implemented the measurement of these predictors in daily postoperative routine. Extensively testing cognitive functions is almost infeasible for every patient during the anesthesia consent meeting. Hence, we only found less than 2% of patients with a present Mini-Cog. Since we extracted routinely collected electronic health records, we could not include these predictors in our feature sets. However, we encoded comorbidities with the help of International Classification of Diseases (ICD) codes.

We added a list of ICD German Modification codes as Table B.5 in supplement B. Please find corresponding codes for cognitive impairment (U51), dementia (F01-F03), chronic pain (R52.1, R52.2), and sleep apnea (G47.3) in addition to other comorbidities that have been identified by previous work [2, 4, 5]. We believe that our models are more generalizable to other institutions since ICD codes are used for billing in virtually all hospitals.

In the **Feature Selection** section, we added “The patient's medical history was encoded by ICD codes, surgical procedures were extracted via the German Operationen und Prozeduren Schlüssel (OPS) (see Table B.5 in Supplement B).”

In the **Limitations** section, we added “Previous clinical studies have highlighted the importance of composite predictors like the Charlson Comorbidity Index (CCI). In our study, selected features relied on routinely collected data omitting these composite variables. Instead, we trained models with the history of International Classification of Diseases (ICD) codes that can be freely combined by our models for addressing the POD prediction task”

[2] Cherak, S. J., Soo, A., Brown, K. N., Ely, E. W., Stelfox, H. T., & Fiest, K. M. (2020). Development and validation of delirium prediction model for critically ill adults parameterized to ICU admission acuity. *PloS one*, 15(8), e0237639.

[3] Yajima, S., Nakanishi, Y., Matsumoto, S., Ookubo, N., Tanabe, K., Kataoka, M., & Masuda, H. (2022). Mini-cog to predict postoperative delirium in patients who underwent transurethral resection of bladder tumor while awake. *Turkish Journal of Urology*, 48(2), 106.

[4] de la Varga-Martínez, O., Gómez-Pesquera, E., Muñoz-Moreno, M. F., Marcos-Vidal, J. M., López-Gómez, A., Rodenas-Gómez, F., ... & Gómez-Sánchez, E. (2021). Development and validation of a delirium risk prediction preoperative model for cardiac surgery patients (DELIPRECA): An observational multicentre study. *Journal of Clinical Anesthesia*, 69, 110158.

[5] Fan, H., Ji, M., Huang, J., Yue, P., Yang, X., Wang, C., & Ying, W. (2019). Development and validation of a dynamic delirium prediction rule in patients admitted to the Intensive Care Units (DYNAMIC-ICU): A prospective cohort study. *International journal of nursing studies*, 93, 64-73.

COMMENT #3: Of specific interest, is the evidence on the relationship between arterial blood pressure and POD that several studies do not confirm as a POD risk factor. Is not clear if – according the Authors- the presented results should replace or complete the available evidence.

RESPONSE #3: Thank you very much for pointing out that arterial blood pressure (APB) was not strongly confirmed as a POD predictor by previous studies. In principle, we agree with this observation [6]. However, we found studies that identified intraoperative fluctuations of ABP as discriminative input variables:

We added “Hirsch et al. [7] found that variance in systolic blood pressure had significant effects on POD. Wang et al. [8] reported that both high and low mean blood pressure levels (deviating from 80 mmHg) were significantly associated with POD occurrence” to the section **Clinical Implications**.

In our study, we describe changes of ABP levels in more detail by including univariate tests as mixed linear effect models and Spearman correlations for distinct intraoperative time-windows. Additionally, multivariable temporal models, such as LSTM, easily learn from variances in time series. Our results comprise significant effects of increased levels of non-invasive - (p-value: 1.248e-26) as well as invasive systolic blood pressure (p-value: 2.38e-13) measured during the surgery. We see these findings as proxies for increased surgical stress that is proven to be associated with POD [9]. On the contrary, Xu et al. claim that actively maintaining high intraoperative ABP has a significant positive effect on outcomes [10].

We formulated “We found high levels of blood pressure towards the end of the surgery and reduced doses of the vasoconstructive drug norepinephrine to be positively correlated with POD. (...) Elevated blood pressure might indicate surgical stress evidently associated with POD [9]. (...) Xue et al. suggest actively maintaining high blood pressure values for improving POD outcomes of patients undergoing hip-replacements [10]. (...) We see our results as complementary with previous studies but we stress the need for taking temporal intraoperative dynamics into account when predicting clinical outcomes.” in **Clinical Implications**.

Additionally, we investigated the predictive power of missing indicators for blood pressure values, like invasive systolic ABP (ibp_sys), as “The presence of invasive ibp_sys measurements can be seen as a proxy for more serious disease but also for the invasiveness or extent of the surgery” (**Identified Predictors** section). We think that future studies on routinely measured health records can benefit from these findings.

[6] Sakaguchi, T., Watanabe, M., Kawasaki, C., Kuroda, I., Abe, H., Date, M., ... & Koretsune, Y. (2018). A novel scoring system to predict delirium and its relationship with the clinical course in patients with acute decompensated heart failure. *Journal of cardiology*, 71(6), 564-569.

[7] Hirsch, J., DePalma, G., Tsai, T. T., Sands, L. P., & Leung, J. M. (2015). Impact of intraoperative hypotension and blood pressure fluctuations on early postoperative delirium after non-cardiac surgery. *British journal of anaesthesia*, 115(3), 418-426.

[8] Wang, N. Y., Hirao, A., & Sieber, F. (2015). Association between intraoperative blood pressure and postoperative delirium in elderly hip fracture patients. *PloS one*, 10(4), e0123892.

[9] Aldecoa, C., Bettelli, G., Bilotta, F., Sanders, R. D., Audisio, R., Borozdina, A., ... & Spies, C. D. (2017). European Society of Anaesthesiology evidence-based and consensus-based guideline on postoperative delirium. *European Journal of Anaesthesiology| EJA*, 34(4), 192-214.

[10] Xu, X., Hu, X., Wu, Y., Li, Y., Zhang, Y., Zhang, M., & Yang, Q. (2020). Effects of different BP management strategies on postoperative delirium in elderly patients undergoing hip replacement: a single center randomized controlled trial. *Journal of clinical anesthesia*, 62, 109730.

COMMENT #4: Inadequate screening for relevant variables (preoperative Comorbidity and cognitive function, etc) might include a bias that possibly prevent to confirm the presented association.

RESPONSE #4: We have explained our feature encodings for decreased cognitive functions and additional comorbidities in our RESPONSE #2. We do not claim that presented associations were proven to be causal due to the highly complex interactions between clinical parameters.

We formulated “Results show that clinical variables are highly dependent on each other and that we cannot differentiate between treatment indication and treatment effect.” (in **Clinical Implications**)

We added “We observed that trained models benefited from the inclusion of multiple inter-correlated time series. We did not apply advanced mathematical methods, like causal inference models [11], to correct for confounding. However, we believe that the inclusion of features that are not necessarily associated with our target variable, such as suppressor variables [12], potentially increased the predictive power of our models” to the **Limitations** section.

[11] Prosperi, M., Guo, Y., Sperrin, M., Koopman, J. S., Min, J. S., He, X., ... & Bian, J. (2020). Causal inference and counterfactual prediction in machine learning for actionable healthcare. *Nature Machine Intelligence*, 2(7), 369-375.

[12] Wilming, R., Kieslich, L., Clark, B., & Haufe, S. (2023, July). Theoretical behavior of XAI methods in the presence of suppressor variables. In *International Conference on Machine Learning* (pp. 37091-37107). PMLR.

COMMENT #5: Is not clear if the Authors consider that the reported accuracy (AUC 0.7-0.8) is adequate to design specific clinical decision for the subgroup of patients at “higher risk” in a prospective study.

RESPONSE #5: Thank you for requesting more interpretation about the clinical application of our models. In the section **Scenario Analysis**, we added three potential scenarios for a model configuration finding a balance between patient safety and overtreatment. We wrote “Assuming 100 surgeries per day, we calculated confusion matrices for three different threshold configurations either A) maximizing sensitivity and specificity, B) maximizing precision and recall, or C) setting recall to 0.8 (see Figure 7). In A), the model configuration led to a sensitivity and specificity both at 0.711. Scenario B) achieved the highest recall out of all scenarios with 0.974 leading to 8 True Positives (TP) out of 9 Positives at the cost of 66 False Positives (FP). The number of FP could be decreased to 36 in C) at 0.163 precision.”. We are aware of potentially increasing short-term hospitalization costs when dealing with elevated false positive rates, like in scenario C). We address these in **Limitations** by “However, especially overtreatment that arises for low prevalences, such as 8.27% here, could hinder implementation of TRAPOD in clinical practice”.

We believe that, with respect to the complexity of our prediction problem, our findings can contribute to a prospective study identifying high-risk patients. In the section **Outlook**, we added “Future prospective studies will directly assess the added value of TRAPOD for clinical decision support. Hereby, we hope that the clinical relevance of TRAPOD can be validated in practice facing obstacles like delayed - or bad data, low prevalence rates, or flawed documentation, that are observed in routinely collected health records”. A dashboard solution that predicts on live data from our source systems is currently under development and shall be validated by a prospective study.

COMMENT #6: The text is presented as explanation of figures. Rather the Authors should frame the content to deliver to the readers and use the figures to clarify the information. Sentences should –generally- not begin with a “Figure” rather should end (where appropriate) referring to a figure.

RESPONSE #6: Thank you for suggesting that sentences should explain results rather than starting with “Figure”. According to your comment, we have changed the following sentences in our manuscript:

We describe results for the top 20 features whose mean values are most strongly correlated with POD during the entire surgery (see Figure 1).

We calculated correlation coefficients for time-static features being constant during the perioperative phase (see Table A.2 of Supplement A).

We present model coefficients related to fixed effects for the 20 features with largest absolute coefficients for factor target as estimated via separate MLEMs (see Figure 2).

We evaluated the receiver operating characteristic and precision-recall curves from which AUROC and AUPRC metrics are derived (see Figure 3).

We analyzed distributions of AUROC and AUPRC scores retrieved from the bootstrapped test set across model variants (see Figure 4).

We trained combined model architectures comprised of one MLP model and one temporal deep learning model (LSTM or our proposed transformer) acting on time-static as well as time-dynamic data, respectively (see Figure 9). ...

COMMENT #7: Some of the abbreviations are repeated in the text after the first use. Page 2: long short-term memory (LSTM), Page 6: long short-term memory (LSTM) Similarly, when an abbreviation is presented , it should be consistently used thereafter Page 2: Multiple studies used non-linear machine learning (ML) Page 7: machine learning Authors have the responsibility for checking the manuscript accuracy.

RESPONSE #8: Thank you for suggesting that abbreviations should be consequently used after their first introduction. We screened our manuscript for these occurrences and now use abbreviations like nursing delirium screening scale (Nu-DESC), TRAN (transformer), MLP (multi-layer perceptron), LSTM (long short-term memory), ML (machine learning) or DL (deep learning) coherently.

We rephrased sentences like:

The current development of large language models (LLMs) has been driven by the invention of the transformer (TRAN) architecture, which is based on the attention mechanism.

We trained and applied multiple ML model variants comprising stand-alone long LSTM, TRAN, MLP architectures as well as combined models for POD predictions
Presence of POD was labeled with the Nu-DESC.

Missing observations in time series and data heterogeneity have, however, been identified as impeding temporal representation in complex DL models.

Reviewer #2 (Remarks to the Author):

>

This paper explores transformers to learn temporal information in dynamic features to predict post-operative delirium. The paper concludes that the developed model, TRAPOD, outperformed other models. The major weaknesses are that the paper's contribution lacks novelty and needs more baselines for comparison.

>

> Strengthes:

>

> 1) It aims to solve a clinically meaningful problem with real patient data. The developed TRAPOD model, based on the transformer architecture, achieves notable improvements in performance against the baselines included.

>

> 2) It evaluates the model using temporal information from different time intervals prior to or during the intraoperative period for early detection. The statistical analysis is detailed and properly formulated, potentially providing insights into model explanation using attention weights.

>

> 3) Overall, the paper is well-structured and easy to follow.

>

> Weaknesses:

>

COMMENT #8: The evaluation needs to include more experiments to compare TRAPOD with other models. For example, XGBoost has shown robust performance on EHR datasets in previous studies. Among deep learning models, CNN and other variations of recurrent neural network such as GRU-D should be included in the performance comparison.

RESPONSE #8: Thank you for suggesting additional machine learning variants. We agree that tree-based models, like gradient boosted decision trees, have shown robust performances on various clinical prediction tasks. We have evaluated such tree-based models in our previous study [13] for the same prediction setting as present in the TRAPOD paper. We see that TRAPOD especially improves precision performances over these models when trained on intraoperative time series.

Similar to Yu et al. [14], we see CNN well suited for the field of medical image processing applying convolutions to highly dimensional data while learning spatial information. Due to the focus of our paper on learning from temporal dynamics, we think that LSTM and transformer (TRAN) are more sufficient for our POD prediction setting since they are explicitly designed for processing time series [14]. Especially the attention-based TRAN architecture has revealed valuable insights into temporal dynamics by providing attention weights that we analyzed. We added “We acknowledge that additional model architectures, such as CNN-based models [15, 16], could potentially be configured for predictions on clinical time series data. However, we believe that LSTM and TRAN models are well-suited to capture temporal dynamics for learning our POD prediction task.” to the **Limitations** section.

We found the suggested GRU-D model particularly interesting due to its ability to learn from missingness patterns extending the traditional last observation carried forward (LOCF) approach. We implemented, configured, and trained the modified gated recurrent unit. Afterwards, we included baseline results in our manuscript and in the supplement. We introduce the model variant in our **Introduction** by “Che et al. [17] enhanced a gated recurrent unit (GRU) architecture for learning temporal missingness patterns while predicting a clinical target variable.”

Figure A11. Model performance of GRU-D trained on the first intraoperative 30 minutes ([T_begin,30]) time series. Performance was evaluated with AUROC (left graph), and AUPRC (right graph) curves. We display performance of TRAPOD and our best MLP model trained with summary statistics (MLP_TAB_P). Random classification levels were at 0.5 and 0.09 for AUROC and AUPRC, respectively

We enhanced our **Results** section by “In addition to simple open clinical prognostic models, we trained GRU-D on data from window ([T_begin, 30]). The model explicitly learns so-called decay rates describing more advanced missingness patterns than last observation carried forward (LOCF) that is commonly used for imputing unobserved values in multivariable time series. GRU-D achieved a mean AUROC of 0.747 (95%-CI [0.731, 0.751]) and a mean AUPRC of 0.282 (95%-CI [0.271, 0.293]) (see Figure A.11 in Supplement A). These performances line up between our best performing MLP (mean AUROC 0.717 95%-CI [0.715, 0.718]), mean AUPRC 0.221 95%-CI [0.219, 0.223]) and TRAPOD (mean AUROC 0.774 95%-CI [0.772, 0.787], mean AUPRC 0.221 95%-CI [0.328, 0.340]), similar to combined model variants (see Table 1). Analyzing decay rates with high temporal signal for input variables highlighted features that were also identified by previous univariate tests (see Figure A.12 in Supplement A).”

In the section **GRU-D Baseline Model** as part of Supplement A, we explain the mathematical concepts of GRU-D, how we trained the model architecture, and outline validation results. Additionally, we show hidden- as well as input decay rates that can be compared with previous findings drawn by univariate test statistics. We provide our code for implementing GRU-D on GitHub [18]. Our results suggest that GRU-D was successfully trained and learned from encoded multi-dimensional missingness information. Nevertheless, in our scenario, TRAPOD achieved best performance metrics. We sincerely thank the reviewer for suggesting this model variant, which has benefited our submission.

[13] Giesa, N., Haufe, S., Menk, M., Weiß, B., Spies, C. D., Piper, S. K., ... & Boie, S. D. (2024). Predicting postoperative delirium assessed by the Nursing Screening Delirium Scale in the recovery room for non-cardiac surgeries without craniotomy: A retrospective study using a machine learning approach. *PLOS Digital Health*, 3(8), e0000414.

[14] Yu, Z., Wang, K., Wan, Z., Xie, S., & Lv, Z. (2023). Popular deep learning algorithms for disease prediction: a review. *Cluster Computing*, 26(2), 1231-1251.

[15] Wu, H., Hu, T., Liu, Y., Zhou, H., Wang, J., & Long, M. (2022). Timesnet: Temporal 2d-variation modeling for general time series analysis. *arXiv preprint arXiv:2210.02186*.

[16] Bednarski, B. P., Singh, A. D., Zhang, W., Jones, W. M., Naeim, A., & Ramezani, R. (2022). Temporal convolutional networks and data rebalancing for clinical length of stay and mortality prediction. *Scientific Reports*, 12(1), 21247.

[17] Che, Z., Purushotham, S., Cho, K., Sontag, D., & Liu, Y. (2018). Recurrent neural networks for multivariate time series with missing values. *Scientific reports*, 8(1), 6085.

[18] GitHub, https://github.com/ngiesa/TRAPOD/tree/main/baseline_models/gru_d_baseline, accessed 08/21/24.

COMMENT #9: The work does not have enough novelty to inspire future developments in the field. The utilization of a transformer encoder combined with MLP to learn dynamic and static data representations is a well-established approach to multi-modal clinical data, and the conclusion that it outperformed LSTM and MLP is not surprising. The data preprocessing including imputation is quite standard as well. While the authors claim novelty in its encoder-only structure, the contribution is not sufficient.

>

RESPONSE #9: We believe that our main contribution lays in the systematic investigation of temporal dynamics in intraoperative clinical time series. To our knowledge, we are the first group investigating dynamics in intraoperative time series associated with POD. While previous studies have detected associations between aggregated clinical parameters and POD, we could successfully highlight the importance of temporal fluctuations.

Our transformer architecture TRAPOD was constructed with standard components but tailored for our classification task. We did not intend to claim novelty for the architecture since we did not validate its performance further. For making these points clearer to the reader, we altered the manuscript as follows.

We changed the title of our manuscript to “TRAPOD: Exploiting Intraoperative Temporal Dynamics Improves the Prediction of Postoperative Delirium”. In the **Introduction**, we formulated our objectives like “Characteristic temporal relationships between clinical parameters as well as the presence or absence of parameter values might play a crucial role for the POD prediction task. We are not aware of any studies systematically investigating temporal relationships in intraoperative clinical time series”. In the **Limitations** section, we highlight “At this point, it is unclear whether the TRAPOD architecture generalizes to other clinical predictions tasks and settings. Future work will assess its efficacy for additional clinical predictions and validate it in multi-centric studies.”

We apologize if the impression arose that we had developed a completely new architecture that is generalizable for all medical prediction tasks.

Responses to referee's expertise

Referee expertise:

- > Referee #1: clinician, post-operative delirium
- > Referee #2: machine learning

Reviewers' comments:

Reviewer #1 (Remarks to the Author):

- > This retrospective study in more than 60.000 patients is intended to investigate the temporal dynamics in intraoperative parameters that are associated with prediction of POD using a machine learning approach.
- > The proposed approach is interesting and addresses a clinically relevant topic.
- > Despite the clinical relevance and implication, the manuscript needs to be extensively revised for clarity.
- > Furthermore there are some methodological limitations that should be appropriately acknowledged.

Major comments

COMMENT #1: The Authors report that the diagnosis of POD was based on NuDESC. It is important to report the actual length of postoperative monitoring and the daily checking. POD is a fluctuating symptom and accuracy in diagnosis is a key element to frame related risk factors.

RESPONSE #1: Thank you very much for your comment. We completely agree that POD is a fluctuating symptom and reported additional details on the postoperative monitoring and screening procedures.

Specifically, we added "On average, the Nu-DESC assessment was performed 1.2 ± 0.5 (mean \pm std) times, where the first score was assessed 37.39 ± 26.83 minutes after the extubation for the whole patient cohort. Patients experiencing POD had prolonged stays in the hospital (11.45 ± 7.63 vs. 8.31 ± 5.78 days) and in the recovery room (2.51 ± 7.63 vs. 2.03 ± 2.63 hours). They also underwent longer anesthesia (4.65 ± 1.36 vs. 2.99 ± 1.24 hours)." In Table A.1. in Appendix A, we report additional patient characteristics including perioperative durations.

Our target variable relied on the Nu-DESC measured inside the recovery room as our aim was to predict early POD onsets. Hereby, we wanted to help clinicians to identify vulnerable patients that might get transferred to a higher level of care [1]. Nu-DESC scores were assessed by trained nursing staff after the patient regained a certain level of consciousness. Daily follow-up checkups in ICUs were not considered so we focused on a heterogenous patient cohort while coherently defining our POD endpoint.

In the **Limitations** section, we explicitly highlight concerns that you have raised: "Our POD definition relied purely on the Nu-DESC assessment in the recovery room. Hence, we could not prevent the occurrence of possible documentation bias. However, taking Nu-DESC scores measured in the recovery room enabled the investigation of clearly defined prediction time windows that were seen as clinically valuable. (...) We understand that a single assessment score recorded in the recovery room serves as a snapshot and does not reflect POD's fluctuating symptoms over a longer time period. Multiple follow-up assessments may have yielded different outcomes."

[1] Spies, C., Piazena, H., Deja, M., Wernecke, K. D., Willemeit, T., Luetz, A., & ICU Design Working Group. (2024). Modification in ICU design may affect delirium and circadian melatonin: a proof of concept pilot study. *Critical Care Medicine*, 52(4), e182-e192.

COMMENT #2: The variables tested by the Authors are among those routinely monitored in the peri and intraoperative period but do not include some of those that are “proven” to relate with POD (check available literature).

RESPONSE #2: Thank you for expressing concerns about the selected variables. We extensively screened available literature. Composite scores, like the Charlson Comorbidity Index (CCI) [2] or dementia-related tests like the Mini-Cog [3] or the Mini-Mental-Status-Test (MMST) [4], were identified as strong predictors by previous clinical studies.

Our clinical institution has not implemented the measurement of these predictors in daily postoperative routine. Extensively testing cognitive functions is almost infeasible for every patient during the anesthesia consent meeting. Hence, we only found less than 2% of patients with a present Mini-Cog. Since we extracted routinely collected electronic health records, we could not include these predictors in our feature sets. However, we encoded comorbidities with the help of International Classification of Diseases (ICD) codes.

We added a list of ICD German Modification codes as Table B.5 in supplement B. Please find corresponding codes for cognitive impairment (U51), dementia (F01-F03), chronic pain (R52.1, R52.2), and sleep apnea (G47.3) in addition to other comorbidities that have been identified by previous work [2, 4, 5]. We believe that our models are more generalizable to other institutions since ICD codes are used for billing in virtually all hospitals.

In the **Feature Selection** section, we added “The patient's medical history was encoded by ICD codes, surgical procedures were extracted via the German Operationen und Prozeduren Schlüssel (OPS) (see Table B.5 in Supplement B).”

In the **Limitations** section, we added “Previous clinical studies have highlighted the importance of composite predictors like the Charlson Comorbidity Index (CCI). In our study, selected features relied on routinely collected data omitting these composite variables. Instead, we trained models with the history of International Classification of Diseases (ICD) codes that can be freely combined by our models for addressing the POD prediction task”

[2] Cherak, S. J., Soo, A., Brown, K. N., Ely, E. W., Stelfox, H. T., & Fiest, K. M. (2020). Development and validation of delirium prediction model for critically ill adults parameterized to ICU admission acuity. *PloS one*, 15(8), e0237639.

[3] Yajima, S., Nakanishi, Y., Matsumoto, S., Ookubo, N., Tanabe, K., Kataoka, M., & Masuda, H. (2022). Mini-cog to predict postoperative delirium in patients who underwent transurethral resection of bladder tumor while awake. *Turkish Journal of Urology*, 48(2), 106.

[4] de la Varga-Martínez, O., Gómez-Pesquera, E., Muñoz-Moreno, M. F., Marcos-Vidal, J. M., López-Gómez, A., Rodenas-Gómez, F., ... & Gómez-Sánchez, E. (2021). Development and validation of a delirium risk prediction preoperative model for cardiac surgery patients (DELIPRECA): An observational multicentre study. *Journal of Clinical Anesthesia*, 69, 110158.

[5] Fan, H., Ji, M., Huang, J., Yue, P., Yang, X., Wang, C., & Ying, W. (2019). Development and validation of a dynamic delirium prediction rule in patients admitted to the Intensive Care Units (DYNAMIC-ICU): A prospective cohort study. *International journal of nursing studies*, 93, 64-73.

COMMENT #3: Of specific interest, is the evidence on the relationship between arterial blood pressure and POD that several studies do not confirm as a POD risk factor. Is not clear if – according the Authors- the presented results should replace or complete the available evidence.

RESPONSE #3: Thank you very much for pointing out that arterial blood pressure (APB) was not strongly confirmed as a POD predictor by previous studies. In principle, we agree with this observation [6]. However, we found studies that identified intraoperative fluctuations of ABP as discriminative input variables:

We added “Hirsch et al. [7] found that variance in systolic blood pressure had significant effects on POD. Wang et al. [8] reported that both high and low mean blood pressure levels (deviating from 80 mmHg) were significantly associated with POD occurrence” to the section **Clinical Implications**.

In our study, we describe changes of ABP levels in more detail by including univariate tests as mixed linear effect models and Spearman correlations for distinct intraoperative time-windows. Additionally, multivariable temporal models, such as LSTM, easily learn from variances in time series. Our results comprise significant effects of increased levels of non-invasive - (p-value: 1.248e-26) as well as invasive systolic blood pressure (p-value: 2.38e-13) measured during the surgery. We see these findings as proxies for increased surgical stress that is proven to be associated with POD [9]. On the contrary, Xu et al. claim that actively maintaining high intraoperative ABP has a significant positive effect on outcomes [10].

We formulated “We found high levels of blood pressure towards the end of the surgery and reduced doses of the vasoconstructive drug norepinephrine to be positively correlated with POD. (...) Elevated blood pressure might indicate surgical stress evidently associated with POD [9]. (...) Xue et al. suggest actively maintaining high blood pressure values for improving POD outcomes of patients undergoing hip-replacements [10]. (...) We see our results as complementary with previous studies but we stress the need for taking temporal intraoperative dynamics into account when predicting clinical outcomes.” in **Clinical Implications**.

Additionally, we investigated the predictive power of missing indicators for blood pressure values, like invasive systolic ABP (ibp_sys), as “The presence of invasive ibp_sys measurements can be seen as a proxy for more serious disease but also for the invasiveness or extent of the surgery” (**Identified Predictors** section). We think that future studies on routinely measured health records can benefit from these findings.

[6] Sakaguchi, T., Watanabe, M., Kawasaki, C., Kuroda, I., Abe, H., Date, M., ... & Koretsune, Y. (2018). A novel scoring system to predict delirium and its relationship with the clinical course in patients with acute decompensated heart failure. *Journal of cardiology*, 71(6), 564-569.

[7] Hirsch, J., DePalma, G., Tsai, T. T., Sands, L. P., & Leung, J. M. (2015). Impact of intraoperative hypotension and blood pressure fluctuations on early postoperative delirium after non-cardiac surgery. *British journal of anaesthesia*, 115(3), 418-426.

[8] Wang, N. Y., Hirao, A., & Sieber, F. (2015). Association between intraoperative blood pressure and postoperative delirium in elderly hip fracture patients. *PLoS one*, 10(4), e0123892.

[9] Aldecoa, C., Bettelli, G., Bilotta, F., Sanders, R. D., Audisio, R., Borzodina, A., ... & Spies, C. D. (2017). European Society of Anaesthesiology evidence-based and consensus-based guideline on postoperative delirium. *European Journal of Anaesthesiology| EJA*, 34(4), 192-214.

[10] Xu, X., Hu, X., Wu, Y., Li, Y., Zhang, Y., Zhang, M., & Yang, Q. (2020). Effects of different BP management strategies on postoperative delirium in elderly patients undergoing hip replacement: a single center randomized controlled trial. *Journal of clinical anesthesia*, 62, 109730.

COMMENT #4: Inadequate screening for relevant variables (preoperative Comorbidity and cognitive function, etc) might include a bias that possibly prevent to confirm the presented association.

RESPONSE #4: We have explained our feature encodings for decreased cognitive functions and additional comorbidities in our RESPONSE #2. We do not claim that presented associations were proven to be causal due to the highly complex interactions between clinical parameters.

We formulated “Results show that clinical variables are highly dependent on each other and that we cannot differentiate between treatment indication and treatment effect.” (in **Clinical Implications**)

We added “We observed that trained models benefited from the inclusion of multiple inter-correlated time series. We did not apply advanced mathematical methods, like causal inference models [11], to correct for confounding. However, we believe that the inclusion of features that are not necessarily associated with our target variable, such as suppressor variables [12], potentially increased the predictive power of our models” to the **Limitations** section.

[11] Prosperi, M., Guo, Y., Sperrin, M., Koopman, J. S., Min, J. S., He, X., ... & Bian, J. (2020). Causal inference and counterfactual prediction in machine learning for actionable healthcare. *Nature Machine Intelligence*, 2(7), 369-375.

[12] Wilming, R., Kieslich, L., Clark, B., & Haufe, S. (2023, July). Theoretical behavior of XAI methods in the presence of suppressor variables. In *International Conference on Machine Learning* (pp. 37091-37107). PMLR.

COMMENT #5: Is not clear if the Authors consider that the reported accuracy (AUC 0.7-0.8) is adequate to design specific clinical decision for the subgroup of patients at “higher risk” in a prospective study.

RESPONSE #5: Thank you for requesting more interpretation about the clinical application of our models. In the section **Scenario Analysis**, we added three potential scenarios for a model configuration finding a balance between patient safety and overtreatment. We wrote “Assuming 100 surgeries per day, we calculated confusion matrices for three different threshold configurations either A) maximizing sensitivity and specificity, B) maximizing precision and recall, or C) setting recall to 0.8 (see Figure 7). In A), the model configuration led to a sensitivity and specificity both at 0.711. Scenario B) achieved the highest recall out of all scenarios with 0.974 leading to 8 True Positives (TP) out of 9 Positives at the cost of 66 False Positives (FP). The number of FP could be decreased to 36 in C) at 0.163 precision.”. We are aware of potentially increasing short-term hospitalization costs when dealing with elevated false positive rates, like in scenario C). We address these in **Limitations** by “However, especially overtreatment that arises for low prevalences, such as 8.27% here, could hinder implementation of TRAPOD in clinical practice”.

We believe that, with respect to the complexity of our prediction problem, our findings can contribute to a prospective study identifying high-risk patients. In the section **Outlook**, we added “Future prospective studies will directly assess the added value of TRAPOD for clinical decision support. Hereby, we hope that the clinical relevance of TRAPOD can be validated in practice facing obstacles like delayed - or bad data, low prevalence rates, or flawed documentation, that are observed in routinely collected health records”. A dashboard solution that predicts on live data from our source systems is currently under development and shall be validated by a prospective study.

COMMENT #6: The text is presented as explanation of figures. Rather the Authors should frame the content to deliver to the readers and use the figures to clarify the information. Sentences should –generally- not begin with a “Figure” rather should end (where appropriate) referring to a figure.

RESPONSE #6: Thank you for suggesting that sentences should explain results rather than starting with “Figure”. According to your comment, we have changed the following sentences in our manuscript:

We describe results for the top 20 features whose mean values are most strongly correlated with POD during the entire surgery (see Figure 1).

We calculated correlation coefficients for time-static features being constant during the perioperative phase (see Table A.2 of Supplement A).

We present model coefficients related to fixed effects for the 20 features with largest absolute coefficients for factor target as estimated via separate MLEMs (see Figure 2).

We evaluated the receiver operating characteristic and precision-recall curves from which AUROC and AUPRC metrics are derived (see Figure 3).

We analyzed distributions of AUROC and AUPRC scores retrieved from the bootstrapped test set across model variants (see Figure 4).

We trained combined model architectures comprised of one MLP model and one temporal deep learning model (LSTM or our proposed transformer) acting on time-static as well as time-dynamic data, respectively (see Figure 9). ...

COMMENT #7: Some of the abbreviations are repeated in the text after the first use. Page 2: long short-term memory (LSTM), Page 6: long short-term memory (LSTM) Similarly, when an abbreviation is presented , it should be consistently used thereafter Page 2: Multiple studies used non-linear machine learning (ML) Page 7: machine learning Authors have the responsibility for checking the manuscript accuracy.

RESPONSE #8: Thank you for suggesting that abbreviations should be consequently used after their first introduction. We screened our manuscript for these occurrences and now use abbreviations like nursing delirium screening scale (Nu-DESC), TRAN (transformer), MLP (multi-layer perceptron), LSTM (long short-term memory), ML (machine learning) or DL (deep learning) coherently.

We rephrased sentences like:

The current development of large language models (LLMs) has been driven by the invention of the transformer (TRAN) architecture, which is based on the attention mechanism.

We trained and applied multiple ML model variants comprising stand-alone long LSTM, TRAN, MLP architectures as well as combined models for POD predictions
Presence of POD was labeled with the Nu-DESC.

Missing observations in time series and data heterogeneity have, however, been identified as impeding temporal representation in complex DL models.

Reviewer #2 (Remarks to the Author):

>

This paper explores transformers to learn temporal information in dynamic features to predict post-operative delirium. The paper concludes that the developed model, TRAPOD, outperformed other models. The major weaknesses are that the paper's contribution lacks novelty and needs more baselines for comparison.

>

> Strengthes:

>

> 1) It aims to solve a clinically meaningful problem with real patient data. The developed TRAPOD model, based on the transformer architecture, achieves notable improvements in performance against the baselines included.

>

> 2) It evaluates the model using temporal information from different time intervals prior to or during the intraoperative period for early detection. The statistical analysis is detailed and properly formulated, potentially providing insights into model explanation using attention weights.

>

> 3) Overall, the paper is well-structured and easy to follow.

>

> Weaknesses:

>

COMMENT #8: The evaluation needs to include more experiments to compare TRAPOD with other models. For example, XGBoost has shown robust performance on EHR datasets in previous studies. Among deep learning models, CNN and other variations of recurrent neural network such as GRU-D should be included in the performance comparison.

RESPONSE #8: Thank you for suggesting additional machine learning variants. We agree that tree-based models, like gradient boosted decision trees, have shown robust performances on various clinical prediction tasks. We have evaluated such tree-based models in our previous study [13] for the same prediction setting as present in the TRAPOD paper. We see that TRAPOD especially improves precision performances over these models when trained on intraoperative time series.

Similar to Yu et al. [14], we see CNN well suited for the field of medical image processing applying convolutions to highly dimensional data while learning spatial information. Due to the focus of our paper on learning from temporal dynamics, we think that LSTM and transformer (TRAN) are more sufficient for our POD prediction setting since they are explicitly designed for processing time series [14]. Especially the attention-based TRAN architecture has revealed valuable insights into temporal dynamics by providing attention weights that we analyzed. We added "We acknowledge that additional model architectures, such as CNN-based models [15, 16], could potentially be configured for predictions on clinical time series data. However, we believe that LSTM and TRAN models are well-suited to capture temporal dynamics for learning our POD prediction task." to the **Limitations** section.

We found the suggested GRU-D model particularly interesting due to its ability to learn from missingness patterns extending the traditional last observation carried forward (LOCF) approach. We implemented, configured, and trained the modified gated recurrent unit. Afterwards, we included baseline results in our manuscript and in the supplement. We introduce the model variant in our **Introduction** by "Che et al. [17] enhanced a gated recurrent unit (GRU) architecture for learning temporal missingness patterns while predicting a clinical target variable."

Figure A11. Model performance of GRU-D trained on the first intraoperative 30 minutes ([T_begin,30]) time series. Performance was evaluated with AUROC (left graph), and AUPRC (right graph) curves. We display performance of TRAPOD and our best MLP model trained with summary statistics (MLP_TAB_P). Random classification levels were at 0.5 and 0.09 for AUROC and AUPRC, respectively

We enhanced our **Results** section by “In addition to simple open clinical prognostic models, we trained GRU-D on data from window ([T_begin, 30]). The model explicitly learns so-called decay rates describing more advanced missingness patterns than last observation carried forward (LOCF) that is commonly used for imputing unobserved values in multivariable time series. GRU-D achieved a mean AUROC of 0.747 (95%-CI [0.731, 0.751]) and a mean AUPRC of 0.282 (95%-CI [0.271, 0.293]) (see Figure A.11 in Supplement A). These performances line up between our best performing MLP (mean AUROC 0.717 95%-CI [0.715, 0.718]), mean AUPRC 0.221 95%-CI [0.219, 0.223]) and TRAPOD (mean AUROC 0.774 95%-CI [0.772, 0.787], mean AUPRC 0.221 95%-CI [0.328, 0.340]), similar to combined model variants (see Table 1). Analyzing decay rates with high temporal signal for input variables highlighted features that were also identified by previous univariate tests (see Figure A.12 in Supplement A).”

In the section **GRU-D Baseline Model** as part of Supplement A, we explain the mathematical concepts of GRU-D, how we trained the model architecture, and outline validation results. Additionally, we show hidden- as well as input decay rates that can be compared with previous findings drawn by univariate test statistics. We provide our code for implementing GRU-D on GitHub [18]. Our results suggest that GRU-D was successfully trained and learned from encoded multi-dimensional missingness information. Nevertheless, in our scenario, TRAPOD achieved best performance metrics. We sincerely thank the reviewer for suggesting this model variant, which has benefited our submission.

[13] Giesa, N., Haufe, S., Menk, M., Weiß, B., Spies, C. D., Piper, S. K., ... & Boie, S. D. (2024). Predicting postoperative delirium assessed by the Nursing Screening Delirium Scale in the recovery room for non-cardiac surgeries without craniotomy: A retrospective study using a machine learning approach. *PLOS Digital Health*, 3(8), e0000414.

[14] Yu, Z., Wang, K., Wan, Z., Xie, S., & Lv, Z. (2023). Popular deep learning algorithms for disease prediction: a review. *Cluster Computing*, 26(2), 1231-1251.

[15] Wu, H., Hu, T., Liu, Y., Zhou, H., Wang, J., & Long, M. (2022). Timesnet: Temporal 2d-variation modeling for general time series analysis. *arXiv preprint arXiv:2210.02186*.

[16] Bednarski, B. P., Singh, A. D., Zhang, W., Jones, W. M., Naeim, A., & Ramezani, R. (2022). Temporal convolutional networks and data rebalancing for clinical length of stay and mortality prediction. *Scientific Reports*, 12(1), 21247.

[17] Che, Z., Purushotham, S., Cho, K., Sontag, D., & Liu, Y. (2018). Recurrent neural networks for multivariate time series with missing values. *Scientific reports*, 8(1), 6085.

[18] GitHub, https://github.com/ngiesa/TRAPOD/tree/main/baseline_models/gru_d_baseline, accessed 08/21/24.

COMMENT #9: The work does not have enough novelty to inspire future developments in the field. The utilization of a transformer encoder combined with MLP to learn dynamic and static data representations is a well-established approach to multi-modal clinical data, and the conclusion that it outperformed LSTM and MLP is not surprising. The data preprocessing including imputation is quite standard as well. While the authors claim novelty in its encoder-only structure, the contribution is not sufficient.

>

RESPONSE #9: We believe that our main contribution lays in the systematic investigation of temporal dynamics in intraoperative clinical time series. To our knowledge, we are the first group investigating dynamics in intraoperative time series associated with POD. While previous studies have detected associations between aggregated clinical parameters and POD, we could successfully highlight the importance of temporal fluctuations.

Our transformer architecture TRAPOD was constructed with standard components but tailored for our classification task. We did not intend to claim novelty for the architecture since we did not validate its performance further. For making these points clearer to the reader, we altered the manuscript as follows.

We changed the title of our manuscript to “TRAPOD: Exploiting Intraoperative Temporal Dynamics Improves the Prediction of Postoperative Delirium”. In the **Introduction**, we formulated our objectives like “Characteristic temporal relationships between clinical parameters as well as the presence or absence of parameter values might play a crucial role for the POD prediction task. We are not aware of any studies systematically investigating temporal relationships in intraoperative clinical time series”. In the **Limitations** section, we highlight “At this point, it is unclear whether the TRAPOD architecture generalizes to other clinical predictions tasks and settings. Future work will assess its efficacy for additional clinical predictions and validate it in multi-centric studies.”

We apologize if the impression arose that we had developed a completely new architecture that is generalizable for all medical prediction tasks.